# Unifying network model links recency and central tendency biases in working memory

**Vezha Boboeva[1,2], Alberto Pezzotta[3,4], Claudia Clopath[1,2]\*†, Athena Akrami[1]\*†**

[1]Sainsbury Wellcome Centre, University College London, London, United Kingdom; [2]Department of Bioengineering, Imperial College London, London, United Kingdom; [3]Gatsby Computational Neuroscience Unit, University College London, London, United Kingdom; [4]The Francis Crick Institute, London, United Kingdom

**\*For correspondence:**
c.clopath@imperial.ac.uk (CC);
athena.akrami@ucl.ac.uk (AA)

†These authors jointly supervised this work

**Competing interest:** The authors declare that no competing interests exist.

**Abstract** The central tendency bias, or contraction bias, is a phenomenon where the judgment of the magnitude of items held in working memory appears to be biased toward the average of past observations. It is assumed to be an optimal strategy by the brain and commonly thought of as an expression of the brain's ability to learn the statistical structure of sensory input. On the other hand, recency biases such as serial dependence are also commonly observed and are thought to reflect the content of working memory. Recent results from an auditory delayed comparison task in rats suggest that both biases may be more related than previously thought: when the posterior parietal cortex (PPC) was silenced, both short-term and contraction biases were reduced. By proposing a model of the circuit that may be involved in generating the behavior, we show that a volatile working memory content susceptible to shifting to the past sensory experience – producing short-term sensory history biases – naturally leads to contraction bias. The errors, occurring at the level of individual trials, are sampled from the full distribution of the stimuli and are not due to a gradual shift of the memory toward the sensory distribution's mean. Our results are consistent with a broad set of behavioral findings and provide predictions of performance across different stimulus distributions and timings, delay intervals, as well as neuronal dynamics in putative working memory areas. Finally, we validate our model by performing a set of human psychophysics experiments of an auditory parametric working memory task.

## eLife assessment

This **important** study combines disparate results from both psychophysics and neural silencing experiments to suggest a new interpretation of how animals and humans represent and interpret recent events in our memory. A key aspect of the model put forward here is the presence of discrete jumps in neural activity within the posterior parietal region of the cortex. The model is distinct from other models, and the authors provide **convincing** evidence to support it both from existing results as well as from novel experiments.

## Introduction

A fundamental question in neuroscience relates to how brains efficiently process the statistical regularities of the environment to guide behavior. Exploiting such regularities may be of great value to survival in the natural environment, but may lead to biases in laboratory tasks. Repeatedly observed across species and sensory modalities is the central tendency ('contraction') bias, where performance in perceptual tasks seemingly reflects a shift of the working memory (WM) representation toward the

**eLife digest** During cognitive tasks, our brain needs to temporarily hold and manipulate the information it is processing to decide how best to respond. This ability, known as working memory, is influenced by how the brain represents and processes the sensory world around us, which can lead to biases, such as 'central tendency'.

Consider an experiment where you are presented with a metal bar and asked to recall how long it was after a few seconds. Typically, our memories, averaged over many trials of repeating this memory recall test, appear to skew towards an average length, leading to the tendency to mis-remember the bar as being shorter or longer than it actually was. This central tendency occurs in most species, and is thought to be the result of the brain learning which sensory input is the most likely to occur out of the range of possibilities.

Working memory is also influenced by short-term history or recency bias, where a recent past experience influences a current memory. Studies have shown that 'turning off' a region of the rat brain called the posterior parietal cortex removes the effects of both recency bias and central tendency on working memory. Here, Boboeva et al. reveal that these two biases, which were thought to be controlled by separate mechanisms, may in fact be related.

Building on the inactivation study, the team modelled a circuit of neurons that can give rise to the results observed in the rat experiments, as well as behavioural results in humans and primates. The computational model contained two modules: one of which represented a putative working memory, and another which represented the posterior parietal cortex which relays sensory information about past experiences.

Boboeva et al. found that sensory inputs relayed from the posterior parietal cortex module led to recency biases in working memory. As a result, central tendency naturally emerged without needing to add assumptions to the model about which sensory input is the most likely to occur. The computational model was also able to replicate all known previous experimental findings, and made some predictions that were tested and confirmed by psychophysics tests on human participants.

The findings of Boboeva et al. provide a new potential mechanism for how central tendency emerges in working memory. The model suggests that to achieve central tendency prior knowledge of how a sensory stimulus is distributed in an environment is not required, as it naturally emerges due to a volatile working memory which is susceptible to errors. This is the first mechanistic model to unify these two sources of bias in working memory. In the future, this could help advance our understanding of certain psychiatric conditions in which working memory and sensory learning are impaired.

---

mean of the sensory history (*Hollingworth, 1910*; *Jou et al., 2004*; *Berliner et al., 1977*; *Hellström, 1985*; *Raviv et al., 2012*; *Fischer and Whitney, 2014*). Equally common are sequential biases, either attractive or repulsive, toward the immediate sensory history (*Akrami et al., 2018*; *Raviv et al., 2012*; *Kiyonaga et al., 2017*; *Cicchini et al., 2017*; *Czoschke et al., 2019*; *Alais et al., 2018*; *Manassi et al., 2018*; *Manassi et al., 2017*; *Suárez-Pinilla et al., 2018*; *Fischer and Whitney, 2014*; *Papadimitriou et al., 2015*).

It is commonly thought that these biases occur due to disparate mechanisms – contraction bias is widely thought to be a result of learning the statistical structure of the environment, whereas serial biases are thought to reflect the contents of WM (*Lieder et al., 2019*; *Barbosa and Compte, 2020*). Recent evidence, however, challenges this picture: our recent study of a parametric working memory (PWM) task discovered that the rat posterior parietal cortex (PPC) plays a key role in modulating contraction bias (*Akrami et al., 2018*). When the region is optogenetically inactivated, contraction bias is attenuated. Intriguingly, however, this is also accompanied by the suppression of bias effects induced by the recent history of the stimuli, suggesting that the two phenomena may be interrelated. Interestingly, other behavioral components, including WM of immediate sensory stimuli (in the current trial), remain intact. In another recent study with humans, a double dissociation was reported between three cohorts of subjects: subjects on the autistic spectrum (ASD) expressed reduced biases due to recent statistics, whereas dyslexic subjects (DYS) expressed reduced biases toward long-term statistics, relative to neurotypical subjects (NT) (*Lieder et al., 2019*). Finally, further complicating the picture is the observation of not only attractive serial dependency, but also repulsive biases (*Fritsche*

*and Spaak, 2020*). It is as of yet unclear how such biases occur and what mechanisms underlie such history dependencies.

These findings stimulate the question of whether contraction bias and the different types of serial biases are actually related, and if so, how. Although normative models have been proposed to explain these effects (*Ashourian and Loewenstein, 2011*; *Fritsche and Spaak, 2020*; *Lieder et al., 2019*), the neural mechanisms and circuits underlying them remain poorly understood. We address this question through a model of the putative circuit involved in giving rise to the behavior observed in *Akrami et al., 2018*. Our model consists of two continuous (bump) attractor sub-networks, representing a WM module and the PPC. Given the finding that PPC neurons carry more information about stimuli presented during previous trials, the PPC module integrates inputs over a longer timescale relative to the WM network and incorporates firing rate adaptation.

We find that both contraction bias and short-term sensory history effects emerge in the WM network as a result of inputs from the PPC network. Importantly, we see that these effects do not necessarily occur due to separate mechanisms. Rather, in our model, contraction bias emerges as a statistical effect of errors in WM, occurring due to the persisting memory of stimuli shown in the preceding trials. The integration of this persisting memory in the WM module competes with that of the stimulus in the current trial, giving rise to short-term history effects. We conclude that contraction biases in such paradigms may not necessarily reflect explicit learning of regularities or an 'attraction toward the mean' on individual trials. Rather, it may be an effect emerging at the level of average performance, when in each trial errors are made according to the recent sensory experiences whose distribution follow that of the input stimuli. Furthermore, using the same model, we also show that the biases toward long-term (short-term) statistics inferred from the performance of ASD (DYS) subjects (*Lieder et al., 2019*) may actually reflect short-term biases extending more or less into the past with respect to NT subjects, challenging the hypothesis of a double-dissociation mechanism. Last, we show that as a result of neuronal integration of inputs and adaptation, in addition to attraction effects occurring on a short timescale, repulsion effects are observed on a longer timescale (*Fritsche and Spaak, 2020*).

We make specific predictions on neuronal dynamics in the PPC and downstream WM areas, as well as how contraction bias may be altered, upon manipulations of the sensory stimulus distribution, intertrial and interstimulus delay intervals. We show agreements between the model and our previous results in humans and rats. Finally, we test our model predictions by performing new human auditory PWM tasks. The data is in agreement with our model and not with an alternative Bayesian model.

## Results

### The PPC as a slower integrator network

PWM tasks involve the sequential comparison of two graded stimuli that differ along a physical dimension and are separated by a delay interval of a few seconds (*Figure 1A and B*; *Romo and Salinas, 2003*; *Akrami et al., 2018*; *Ashourian and Loewenstein, 2011*). A key feature emerging from these studies is contraction bias, where the averaged performance is as if the memory of the first stimulus progressively shifts toward the center of a prior distribution built from past sensory history (*Figure 1C*). Additionally, biases toward the most recent sensory stimuli (immediately preceding trials) have also been documented (*Akrami et al., 2018*; *Raviv et al., 2012*).

In order to investigate the circuit mechanisms by which such biases may occur, we use two identical one-dimensional continuous attractor networks to model WM and PPC modules. Neurons are arranged according to their preferential firing locations in a continuous stimulus space, representing the amplitude of auditory stimuli. Excitatory recurrent connections between neurons are symmetric and a monotonically decreasing function of the distance between the preferential firing fields of neurons, allowing neurons to mutually excite one another; inhibition, instead, is uniform. Together, both allow a localized bump of activity to form and be sustained (*Figure 1D and E*; *Sebastian Seung, 1998*; *Wang, 2001*; *Zhong et al., 2020*; *Spalla et al., 2021*; *Wu and Amari, 2005*; *Wu et al., 2016*; *Fung et al., 2008*; *Fung et al., 2010*; *Trappenberg, 2005*). Both networks have free boundary conditions. Neurons in the WM network receive inputs from neurons in the PPC coding for the same stimulus amplitude (*Figure 1D*). Building on experimental findings (*Murray et al., 2014*; *Siegle et al., 2021*; *Gao et al., 2020*; *Wang et al., 2023*; *Mejías and Wang, 2022*; *Ding et al., 2022*), we designed the PPC network such that it integrates activity over a longer timescale compared to the WM network

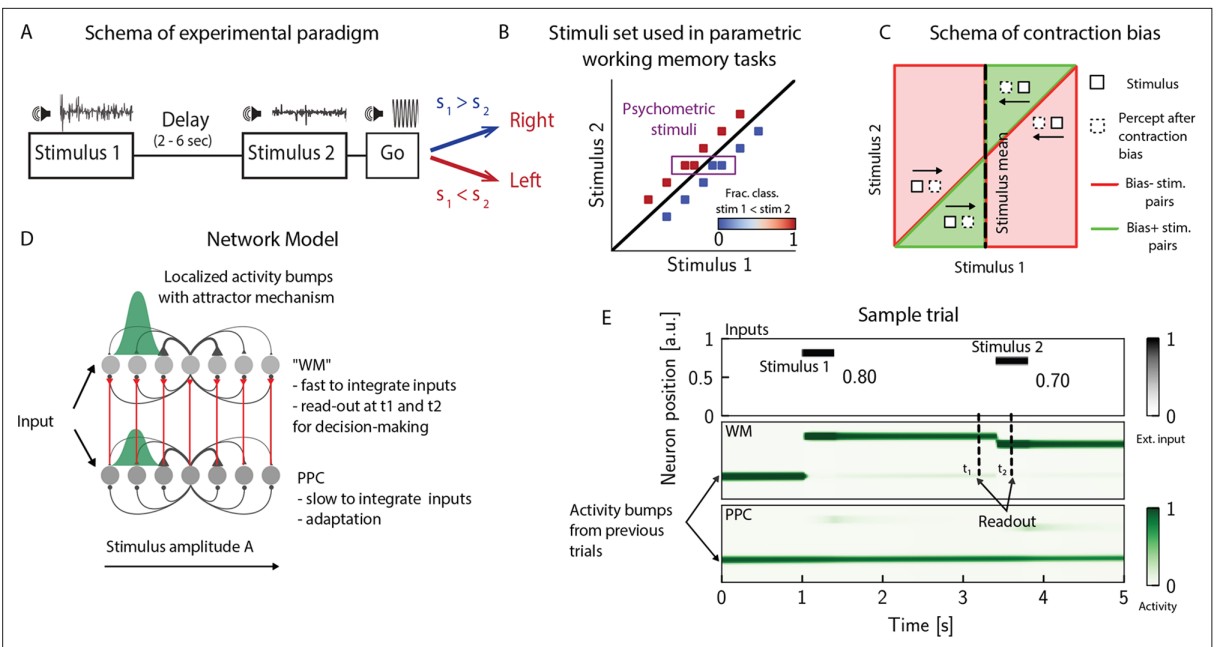

**Figure 1.** The posterior parietal cortex (PPC) as a slower integrator network. (**A**) In any given trial, a pair of stimuli (here, sounds) separated by a variable delay interval is presented to a subject. After the second stimulus, and after a go cue, the subject must decide which of the two sounds is louder by pressing a key (humans) or nose-poking in an appropriate port (rats). (**B**) The stimulus set. The stimuli are linearly separable, and stimulus pairs are equally distant from the $s_1 = s_2$ diagonal. Error-free performance corresponds to network dynamics from which it is possible to classify all the stimuli below the diagonal as $s_1 > s_2$ (shown in blue) and all stimuli above the diagonal as $s_1 < s_2$ (shown in red). An example of a correct trial can be seen in (**E**). In order to assay the psychometric threshold, several additional pairs of stimuli are included (purple box), where the distance to the diagonal $s_1 = s_2$ is systematically changed. The colorbar expresses the fraction classified as $s_1 < s_2$. (**C**) Schematics of contraction bias in delayed comparison tasks. Performance is a function of the difference between the two stimuli, and is impacted by contraction bias, where the base stimulus $s_1$ is perceived as closer to the mean stimulus. This leads to a better/worse (green/red area) performance, depending on whether this 'attraction' increases (Bias+) or decreases (Bias-) the discrimination between the base stimulus $s_1$ and the comparison stimulus $s_2$. (**D**) Our model is composed of two modules, representing working memory (WM) and sensory history (PPC). Each module is a continuous one-dimensional attractor network. Both networks are identical except for the timescales over which they integrate external inputs; PPC has a significantly longer integration timescale and its neurons are additionally equipped with neuronal adaptation. The neurons in the WM network receive input from those in the PPC through connections (red lines) between neurons coding for the same stimulus. Neurons (gray dots) are arranged according to their preferential firing locations. The excitatory recurrent connections between neurons in each network are a symmetric, decreasing function of their preferential firing locations, whereas the inhibitory connections are uniform (black lines). For simplicity, connections are shown for a single presynaptic neuron (where there is a bump in green). When a sufficient amount of input is given to a network, a bump of activity is formed and sustained in the network when the external input is subsequently removed. This activity in the WM network is read out at two time points: slightly before and after the onset of the second stimulus, and is used to assess performance. (**E**) The task involves the comparison of two sequentially presented stimuli, separated by a delay interval (top panel, black lines). The WM network integrates and responds to inputs quickly (middle panel), while the PPC network integrates inputs more slowly (bottom panel). As a result, external inputs (corresponding to stimulus 1 and 2) are enough to displace the bump of activity in the WM network, but not in the PPC. Instead, inputs coming from the PPC into the WM network are not sufficient to displace the activity bump, and the trial is consequently classified as correct. In the PPC, instead, the activity bump corresponds to a stimulus shown in previous trials.

section 'The model'. Moreover, neurons in the PPC are equipped with neural adaptation that can be thought of as a threshold that dynamically follows the activation of a neuron over a longer timescale.

To simulate the PWM task, at the beginning of each trial, the network is provided with a stimulus $s_1$ for a short time via an external current $I_{ext}$ as input to a set of neurons (see **Appendix 1—table 1**). Following $s_1$, after a delay interval, a second stimulus $s_2$ is presented (**Figure 1E**). The pair $(s_1, s_2)$ is drawn from the stimulus set shown in **Figure 1B**, where they are all equally distant from the diagonal $s_1 = s_2$, and are therefore of equal nominal discrimination, or difficulty. The stimuli $(s_1, s_2)$ are co-varied in each trial so that the task cannot be solved by relying on only one of the stimuli (**Hernández et al., 1997**). As in the study in **Akrami et al., 2018** using an interleaved design, across consecutive trials, the interstimulus delay intervals are randomized and sampled uniformly between 2, 6, and 10 s. The intertrial interval (ITI), instead, is fixed at 5 s.

We additionally include psychometric pairs (indicated in the box in *Figure 1B*) where the distance to the diagonal, hence the discrimination difficulty, is varied. The task is a binary comparison task that aims at classifying whether $s_1 < s_2$ or vice versa. In order to solve the task, we record the activity of the WM network at two time points: slightly before and after the onset of $s_2$ (*Figure 1E*). We repeat this procedure across many different trials and use the recorded activity to assess performance (see section 'Simulation') for details. Importantly, at the end of each trial, the activity of both networks is not re-initialized, and the state of the network at the end of the trial serves as the initial network configuration for the next trial.

## Contraction bias and short-term stimulus history effects as a result of PPC network activity

Probing the WM network performance of psychometric stimuli (*Figure 1B*, purple box, 10% of all trials) shows that the comparison behavior is not error-free and that the psychometric curves (different colors) differ from the optimal step function (*Figure 2A*, green dashed line). The performance of pyschometric trials is also better for shorter interstimulus delay intervals, as has been shown in previous work (*Sinclair and Burton, 1996*; *Akrami et al., 2018*). In our model, errors are caused by the displacement of the activity bump in the WM network due to the inputs from the PPC network. These displacements in the WM activity bump can result in different outcomes: by displacing it *away* from the second stimulus, they either do not affect the performance or improve it (*Figure 2B*, right panel, 'Bias+'), if noise is present. Conversely, the performance can suffer, if the displacement of the activity bump is *toward* the second stimulus (*Figure 2B*, left panel, 'Bias-'). Note, however, that in these two specific trials the activity bump in PPC is strong and it displaces the activity bump in the WM network, but this is not the only kind of dynamics present in the network (see section 'Multiple timescales at the core of short-term sensory history effects' for a more detailed analysis of the network dynamics).

Performance of stimulus pairs that are equally distant from the $s_1 = s_2$ diagonal can be similarly impacted and the network produces a pattern of errors that are consistent with contraction bias: performance is at its minimum for stimulus pairs in which $s_1$ is either largest or smallest, and at its maximum for stimulus pairs in which $s_2$ is largest or smallest (*Figure 2C*, left panel; *Ashourian and Loewenstein, 2011*; *Fassihi et al., 2014*; *Akrami et al., 2018*; *Fassihi et al., 2017*; *Esmaeili and Diamond, 2019*). These results are consistent with the performance of humans and rats on the auditory task, as previously reported (*Figure 2C*, middle and right panels, data from *Akrami et al., 2018*).

Can the same circuit also give rise to short-term sensory history biases (*Akrami et al., 2018*; *Loewenstein et al., 2021*)? We analyzed the fraction of trials the network response was '$s_1 < s_2$' in the current trial conditioned on stimulus pairs presented in the previous trial and found that the network behavior is indeed modulated by the preceding trial's stimulus pairs (*Figure 2D*, panel 1). We quantified these history effects as well as how many trials back they extend to. We computed the bias by plotting, for each particular pair (of stimuli) presented at the current trial, the fraction of trials the network response was '$s_1 < s_2$' as a function of the pair presented in the previous trial minus the mean performance over all previous trial pairs (*Figure 2D*, panel 2; *Akrami et al., 2018*). Independent of the current trial, the previous trial exerts an 'attractive' effect, expressed by the negative slope of the line: when the previous pair of stimuli is small, $s_1$ in the current trial is, on average, misclassified as smaller than it actually is, giving rise to the attractive bias in the comparison performance; the converse holds true when the previous pair of stimuli happens to be large. These effects extend to two trials back and are consistent with the performance of humans and rats on the auditory task (*Figure 2D*, panels 3–6, data from *Akrami et al., 2018*).

It has been shown that inactivating the PPC in rats performing the auditory delayed comparison task markedly reduces the magnitude of contraction bias without impacting non-sensory biases (*Akrami et al., 2018*). We assay the causal role of the PPC in generating the sensory history effects as well as contraction bias by weakening the connections from the PPC to the WM network, mimicking the inactivation of the PPC. In this case, we see that the performance of the psychometric stimuli is greatly improved (yellow curve, *Figure 2E*, top panel), consistent also with the inactivation of the PPC in rodents (yellow curve, *Figure 2E*, bottom panel, data from *Akrami et al., 2018*). Performance is improved also for all pairs of stimuli in the stimulus set (*Figure 2—figure supplement 1A*). The breakdown of the network response in the current trial conditioned on the specific stimulus pair preceding

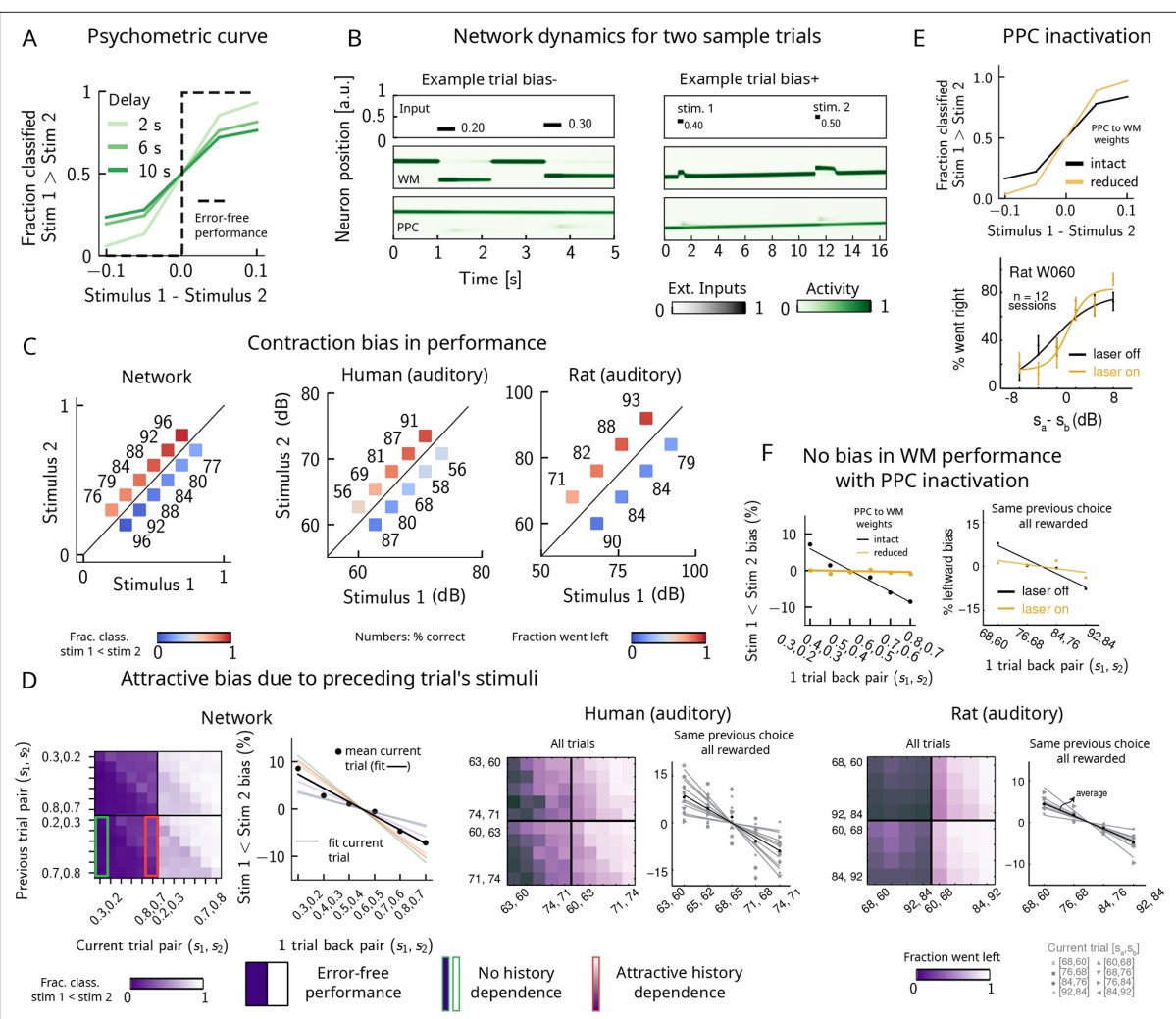

**Figure 2.** Contraction bias and short-term sensory history effects as a result of posterior parietal cortex (PPC) network activity. (**A**) Performance of network model for psychometric stimuli (shades of green) is not error-free (black dashed line). A shorter interstimulus delay interval yields a better performance. (**B**) Errors occur due to the displacement of the bump representing the first stimulus $s_1$ in the working memory (WM) network. Depending on the direction of this displacement with respect to $s_2$, this can give rise to trials in which the comparison task becomes harder (easier), leading to negative (positive) biases (top and bottom panels). Top subpanel: stimuli presented to both networks in time. Middle/bottom subpanels show activity of WM and PPC networks (in green). (**C**) Left: performance is affected by contraction bias – a gradual accumulation of errors for stimuli below (above) the diagonal upon increasing (decreasing) $s_1$. Colorbar indicates fraction of trials classified as $s_1 < s_2$. Middle and right: for comparison, data from the auditory version of the task performed in humans and rats. (**D**) Panel 1: for each combination of current (x-axis) and previous trial's stimulus pair (y-axis), fraction of trials classified as $s_1 < s_2$ (colorbar). Performance is affected by preceding trial's stimulus pair (modulation along the y-axis). For readability, only some tick-labels are shown. Panel 2: bias, quantifying the (attractive) effect of previous stimulus pairs. Colored lines correspond to linear fits of this bias for each pair of stimuli in the current trial. Black dots correspond to average over all current stimuli, and black line is a linear fit. These history effects are attractive: the smaller the previous stimulus, the higher the probability of classifying the first stimulus of the current trial $s_1$ as small, and vice versa. Panel 3: human auditory trials. Percentage of trials in which humans chose left for each combination of current and previous stimuli; vertical modulation indicates attractive effect of preceding trial. Panel 4: percentage of trials in which humans chose left minus the average value of left choices, as a function of the stimuli of the previous trial, for fixed previous trial response choice and reward. Panels 5 and 6: same as panels 3 and 4 but with rat auditory trials. (**E**) Top: performance of network, when the weights from the PPC to the WM network are weakened, is improved for psychometric stimuli (yellow curve), relative to the intact network (black curve). Bottom: psychometric curves for rats (only shown for one rat) are closer to error-free during PPC inactivation (yellow) than during control trials (black). (**F**) Left: the attractive bias due to the effect of the previous trial is present with the default weights (black line), but is eliminated with reduced weights (yellow line). Right: while there is bias induced by previous stimuli in the control experiment (black), this bias is reduced under PPC inactivation (yellow).

*Figure 2 continued*

The online version of this article includes the following figure supplement(s) for figure 2:

**Figure supplement 1.** Inactivating the inputs from the posterior parietal cortex (PPC) network improves performance, in line with experimental findings.

**Figure supplement 2.** Model predictions for a block design.

it reveals that the previous trial no longer exerts a notable modulating effect on the current trial (*Figure 2—figure supplement 1B*). Quantifying this bias by subtracting the mean performance over all of the previous pairs reveals that the attractive bias is virtually eliminated (yellow curve, *Figure 2F*, left panel), consistent with findings in rats (*Figure 2F*, right panel, data from *Akrami et al., 2018*).

Together, our results suggest a possible circuit through which both contraction bias and short-term history effects in a PWM task may arise. The main features of our model are two continuous attractor networks, both integrating the same external inputs, but operating over different timescales. Crucially, the slower one, a model of the PPC, includes neuronal adaptation and provides input to the faster one, intended as a WM circuit. Note that a block design where the delay interval is kept fixed yields similar results (*Figure 2—figure supplement 2*). In the next section, we show how the slow integration and firing rate adaptation in the PPC network give rise to the observed effects of sensory history.

## Multiple timescales at the core of short-term sensory history effects

The activity bumps in the PPC and WM networks undergo different dynamics due to the different timescales with which they integrate inputs, the presence of adaptation in the PPC, and the presence of global inhibition. The WM network integrates inputs over a shorter timescale, and therefore the activity bump follows the external input with high fidelity (*Figure 3A* [purple bumps] and *Figure 3B* [purple line]). The PPC network, instead, has a longer integration timescale, and therefore fails to sufficiently integrate the input to induce a displacement of the bump to the location of a new stimulus, at each single trial. This is mainly due to the competition between the inputs from the recurrent connections sustaining the bump and the external stimuli that are integrated elsewhere: if the former is stronger, the bump is not displaced. If, however, these inputs are weaker, they will not displace it, but may still exert a weakening effect via the global inhibition in the connectivity. The external input, as well as the presence of adaptation (*Figure 3—figure supplement 1B and C*), induces a small continuous drift of the activity bump that is already present from the previous trials (lower-right panel of *Figure 2B*, *Figure 3A* [pink bumps] and *Figure 3B* [pink line]). The build-up of adaptation in the PPC network, combined with the global inhibition from other neurons receiving external inputs, can extinguish the bump in that location (see also *Figure 3—figure supplement 1* for more details). Following this, the PPC network can make a transition to an incoming stimulus position (that may be either $s_1$ or $s_2$), and a new bump is formed. The resulting dynamics in the PPC are a mixture of slow drift over a few trials, followed by occasional jumps (*Figure 3A*).

As a result of such dynamics, relative to the WM network, the activity bump in the PPC represents the stimuli corresponding to the current trial in a smaller fraction of the trials and represents stimuli presented in the previous trial in a larger fraction of the trials (*Figure 3C*). This yields short-term sensory history effects in our model (*Figure 2D and E*) as input from the PPC leads to the displacement of the WM bump to other locations (*Figure 3D*). Given that neurons in the WM network integrate this input, once it has built up sufficiently, it can surpass the self-sustaining inputs from the recurrent connections in the WM network. The WM bump, then, can move to a new location, given by the position of the bump in the PPC (*Figure 3D*). As the input from the PPC builds up gradually, the probability of bump displacement in WM increases over time. This in return leads to an increased probability of contraction bias (*Figure 3E*) and short-term history (one-trial back) biases (*Figure 3F*), as the interstimulus delay interval increases.

Additionally, a non-adapted input from the PPC has a larger likelihood of displacing the WM bump. This is highest immediately following the formation of a new bump in the PPC or, in other words, following a 'bump jump' (*Figure 3F*). As a result, one can reason that those trials immediately following a jump in the PPC are the ones that should yield the maximal bias toward stimuli presented in the previous trial. We therefore separated trials according to whether or not a jump has occurred in the PPC in the preceding trial (we define a jump to have occurred if the bump location across two consecutive trials in the PPC is displaced by an amount larger than the typical width of the bump

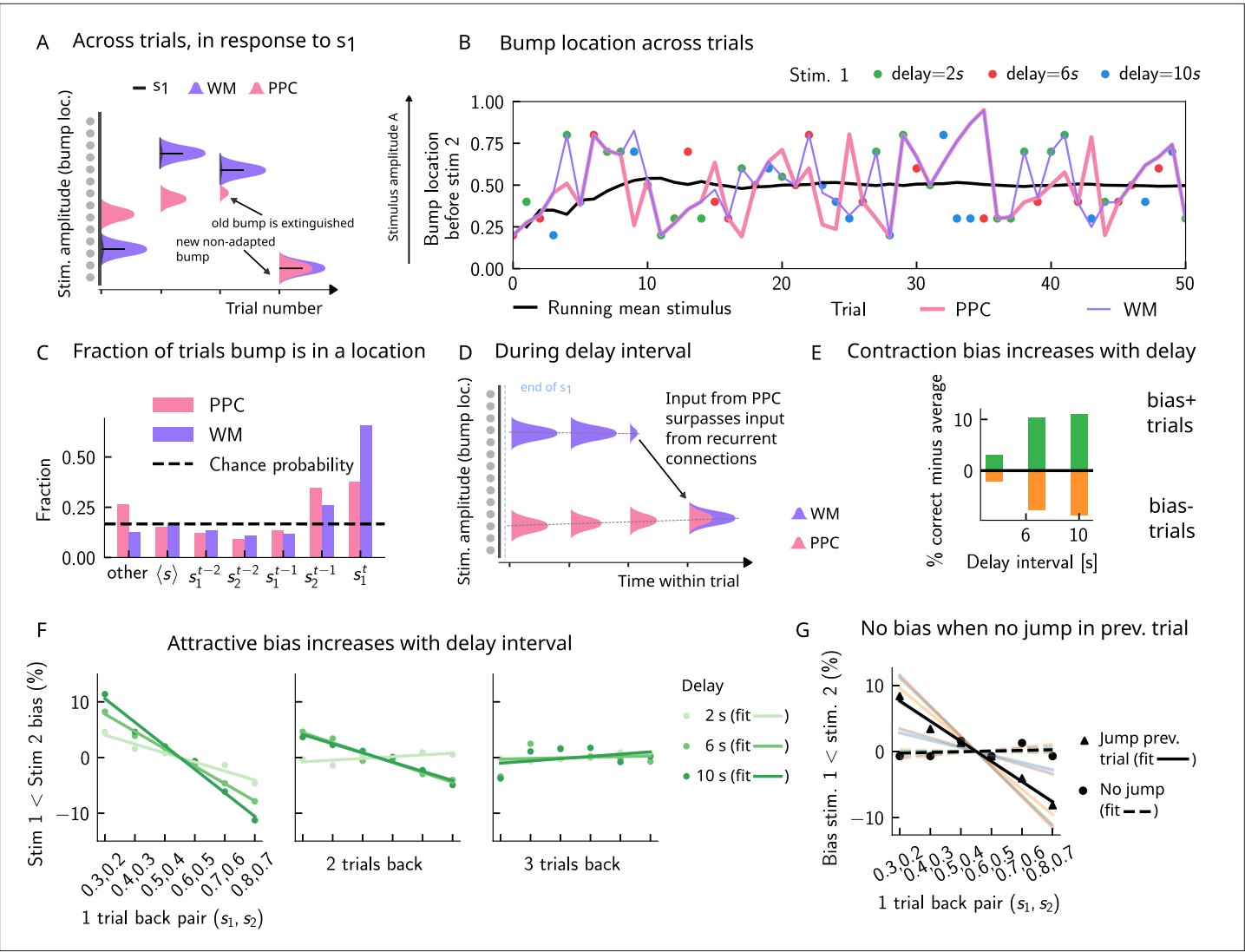

**Figure 3.** Multiple timescales at the core of short-term sensory history effects. (**A**) Schematics of activity bump dynamics in the working memory (WM) vs. posterior parietal cortex (PPC) network. Whereas the WM responds quickly to external inputs, the bump in the PPC drifts slowly and adapts, until it is extinguished and a new bump forms. (**B**) The location of the activity bump in both the PPC (pink line) and the WM (purple line) networks, immediately before the onset of the second stimulus $s_2$ of each trial. This location corresponds to the amplitude of the stimulus being encoded. The bump in the WM network closely represents the stimulus $s_1$ (shown in colored dots, each color corresponding to a different delay interval). The PPC network, instead, being slower to integrate inputs, displays a continuous drift of the activity bump across a few trials before it jumps to a new stimulus location due to the combined effects of inhibition from incoming inputs and adaptation that extinguishes previous activity. (**C**) Fraction of trials in which the bump location corresponds to the base stimulus that has been presented ($s_1^t$) in the current trial, as well as the two preceding trials ($s_2^{t-1}$ to $s_1^{t-2}$). In the WM network, in the majority of trials, the bump coincides with the first stimulus of the current trial $s_1^t$. In a smaller fraction of the trials, it corresponds to the previous stimulus $s_2^{t-1}$ due to the input from the PPC. In the PPC network instead, a smaller fraction of trials consist of the activity bump coinciding with the current stimulus $s_1^t$. Relative to the WM network, the bump is more likely to coincide with the previous trial's comparison stimulus ($s_2^{t-1}$). (**D**) During the interstimulus delay interval, in the absence of external sensory inputs, the activity bump in the WM network is mainly sustained endogenously by the recurrent inputs. It may, however, be destabilized by the continual integration of inputs from the PPC. (**E**) As a result, with an increasing delay interval, given that more errors are made, contraction bias increases. Green (orange) bars correspond to the performance in Bias+ (Bias-) regions, relative to the mean performance over all pairs (*Figure 1C*). (**F**) Left and middle: longer delay intervals allow for a longer integration times, which in turn lead to a larger frequency of WM disruptions due to previous trials, leading to a larger previous trial attractive biases (2 s vs. 6 s vs. 10 s). Right: weak repulsive effects for larger delays become apparent. Colored dots correspond to the bias computed for different values of the interstimulus delay interval, while colored lines correspond to their linear fits. (**G**) When neuronal adaptation is at its lowest in the PPC, that is, following a bump jump, the WM bump is maximally susceptible to inputs from the PPC. The attractive bias (toward previous stimuli) is present in trials in which the PPC network underwent a jump in the previous trial (black triangles, with black line a linear fit). Such biases are absent in trials where no jumps occur in the PPC in the previous

*Figure 3 continued*

trial (black dots, with dashed line a linear fit). Colored lines correspond to bias for specific pairs of stimuli in the current trial, regular lines for the jump condition, and dashed for the no jump condition.

The online version of this article includes the following figure supplement(s) for figure 3:

**Figure supplement 1.** Dynamics of responses in a one-dimensional continuous attractor network in the presence of adaptation.

**Figure supplement 2.** The role of neuronal adaptation in generating short-term history biases.

[section 'The model']). In line with this reasoning, only the set that included trials with jumps in the preceding trial yields a one-trial back bias (*Figure 3G*).

Removing neuronal adaptation entirely from the PPC network further corroborates this result. In this case, the network dynamics show a very different behavior: the activity bump in the PPC undergoes a smooth drift (*Figure 3—figure supplement 2A*), and the bump distribution is much more peaked around the mean (*Figure 3—figure supplement 2B*), relative to when adaptation is present (*Figure 4A*). In this regime, there are no jumps in the PPC (*Figure 3—figure supplement 2A*), and the activity bump corresponds to the stimuli presented in the previous trial in a fewer fraction of the trials (*Figure 3—figure supplement 2C*), relative to when adaptation is present (*Figure 3B*). As a result, no short-term history effects can be observed (*Figure 3—figure supplement 2C and D*), even though a strong contraction bias persists (*Figure 3—figure supplement 2E*).

As in the study in *Akrami et al., 2018*, we can further study the impact of the PPC on the dynamics of the WM network by weakening the weights from the PPC to the WM network, mimicking the inactivation of PPC (*Figure 2E and F*, *Figure 2—figure supplement 1A and B*). Under this manipulation, the trajectory of the activity bump in the WM network immediately before the onset of the second stimulus $s_2$ closely follows the external input, consistent with an enhanced WM function (*Figure 2—figure supplement 1C and D*).

The drift-jump dynamics in our model of the PPC give rise to short-term (notably one- and two-trial back) sensory history effects in the performance of the WM network. In addition, we observe an equally salient contraction bias (bias toward the sensory mean) in the WM network's performance, increasing with the delay period (*Figure 3E*). However, we find that the activity bump in both the WM and the PPC network corresponds to the mean over all stimuli in only a small fraction of trials, expected by chance (*Figure 3B*, see section 'Computing bump location' for how it is calculated). Rather, the bump is located more often at the current trial stimulus ($s_1^t$), and to a lesser extent, at the location of stimuli presented at the previous trial ($s_2^{t-1}$). As a result, contraction bias in our model cannot be attributed to the representation of the running sensory average in the PPC. In the next section, we show how contraction bias arises as an averaged effect when single-trial errors occur due to short-term sensory history biases.

## Errors are drawn from the marginal distribution of stimuli, giving rise to contraction bias

In order to illustrate the statistical origin of contraction bias in our network model, we consider a mathematical scheme of its performance (*Figure 4B*). In this simple formulation, we assume that the first stimulus to be kept in WM, $s_1^t$, is volatile. As a result, in a fraction $\epsilon$ of the trials, it is susceptible to replacement with another stimulus $\hat{s}$ (by the input from the PPC, which has a given distribution $p_m$; *Figure 4A*). However, this replacement does not always lead to an error, as evidenced by Bias- and Bias+ trials (i.e., those trials in which the performance is affected negatively and positively, respectively; *Figure 2B*). For each stimulus pair, the probability to make an error, $p_e$, is integral of $p_m$ over values lying on the wrong side of the $s_1 = s_2$ diagonal (*Figure 4C*). For instance, for stimulus pairs below the diagonal (*Figure 4C*, blue squares) the trial outcome is erroneous only if $\hat{s}$ is displaced above the diagonal (red part of the distribution). As one can see, the area above the diagonal increases as $s_1$ increases, giving rise to a gradual increase in error rates (*Figure 4C*). This mathematical model can capture the performance of the attractor network model, as can be seen through the fit of the network performance, when using the bump distribution in the PPC as $p_m$ and $\epsilon$ as a free parameter (see *Equation 9* in section 'The probability to make errors is proportional to the cumulative distribution of the stimuli, giving rise to contraction bias', *Figure 4D and E*).

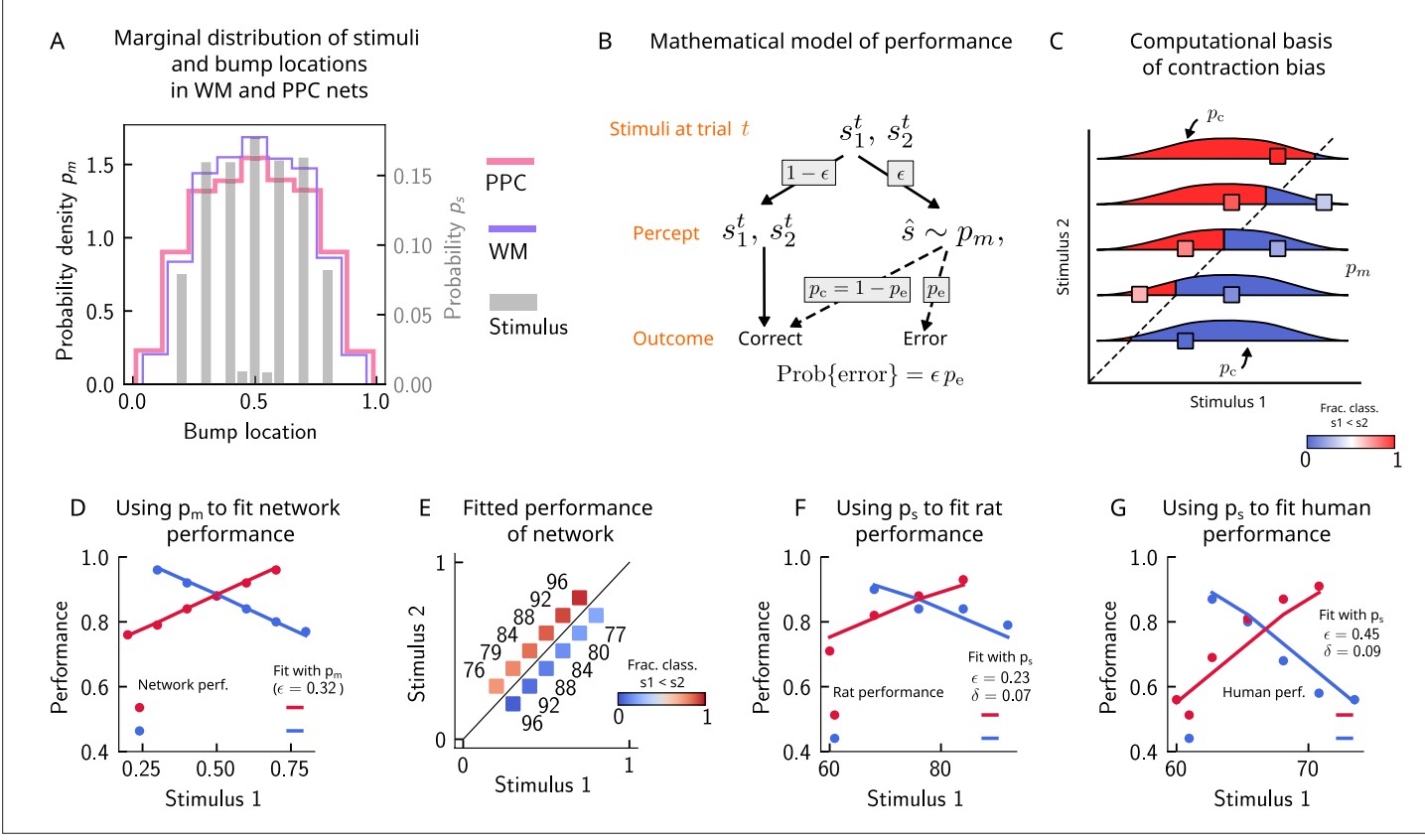

**Figure 4.** Errors are drawn from the marginal distribution of stimuli, giving rise to contraction bias. (**A**) The bump locations in both the working memory (WM) network (in pink) and the posterior parietal cortex (PPC) network (in purple) have identical distributions to that of the input stimulus (marginal over $s_1$ or $s_2$, shown in gray). (**B**) A simple mathematical model illustrates how contraction bias emerges as a result of a volatile working memory for $s_1$. A given trial consists of two stimuli $s_1^t$ and $s_2^t$. We assume that the encoding of the second stimulus $s_2^t$ is error-free, contrary to the first stimulus that is prone to change, with probability $\epsilon$. Furthermore, when $s_1$ does change, it is replaced by another stimulus, $\hat{s}$ (imposed by the input from the PPC in our network model). Therefore, $\hat{s}$ is drawn from the marginal distribution of bump locations in the PPC, which is similar to the marginal stimulus distribution (see panel **B**), $p_m$ (see also section 'The probability to make errors is proportional to the cumulative distribution of the stimuli, giving rise to contraction bias'). Depending on the new location of $\hat{s}$, the comparison to $s_2$ can either lead to an erroneous choice (Bias-, with probability $p_e$) or a correct one (Bias+, with probability $p_c = 1 - p_e$). (**C**) The distribution of bump locations in PPC (from which replacements $\hat{s}$ are sampled) is overlaid on the stimulus set and repeated for each value of $s_2$. For pairs below the diagonal, where $s_1 > s_2$ (blue squares), the trial outcome will be an error if the displaced WM bump $\hat{s}$ ends up above the diagonal (red section of the $p_m$ distribution). The probability to make an error, $p_e$, equals the integral of $p_m$ over values above the diagonal (red part), which increases as $s_1$ increases. Vice versa, for pairs above the diagonal ($s_1 < s_2$, red squares), $p_e$ equals the integral of $p_m$ over values below the diagonal, which increases as $s_1$ decreases. (**D**) The performance of the attractor network as a function of the first stimulus $s_1$, in red dots for pairs of stimuli where $s_1 > s_2$, and in blue dots for pairs of stimuli where $s_1 < s_2$. The solid lines are fits of the performance of the network using *Equation 9*, with $\epsilon$ as a free parameter. (**E**) Numbers correspond to the performance, same as in (**D**), while colors express the fraction classified as $s_1 < s_2$ (colorbar), to illustrate the contraction bias. (**F**) Performance of rats performing the auditory delayed comparison task in *Akrami et al., 2018*. Dots correspond to the empirical data, while the lines are fits with the statistical model, using the distribution of stimuli. The additional parameter $\delta$ captures the lapse rate. (**G**) Same as (**F**), but with humans performing the task. Data in (**F**) and (**G**) reproduced with permission from *Akrami et al., 2018*.

The online version of this article includes the following figure supplement(s) for figure 4:

**Figure supplement 1.** The stimulus distribution impacts the pattern of contraction bias.

Can this simple statistical model also capture the behavior of rats and humans (*Figure 2C*)? We carried out the same analysis for rats and humans by replacing the bump location distribution of PPC with that of the marginal distribution of the stimuli provided in the task based on the observation that the former is well-approximated by the latter (*Figure 4A*). In this case, we see that the model roughly captures the empirical data (*Figure 4F and G*), with the addition of another parameter $\delta$ that accounts for the lapse rate. Interestingly, such 'lapse' also occurs in the network model (as seen by the small amount of errors for pairs of stimuli where $s_2$ is smallest and largest; *Figure 4E*). This occurs because of

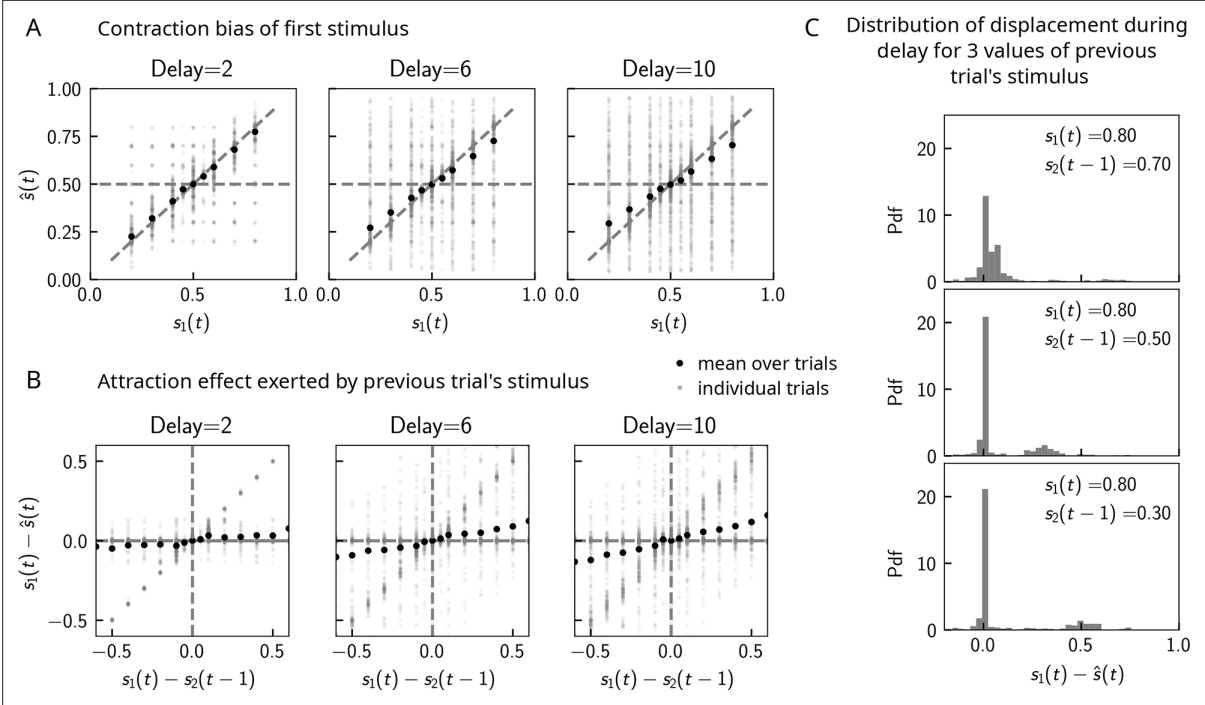

**Figure 5.** Contraction bias in continuous recall. (**A**) We observe contraction bias of the bump of activity after the delay period $\hat{s}$: the average $\hat{s}$ over trials (black dots) deviates from the identity line (diagonal dashed line) toward the mean of the marginal stimulus distribution (0.5). This effect is stronger as the delay interval is longer (left to right panel). (**B**) This contraction bias is actually largely due to the effect of the previous trial: the larger the difference between the current trial and the previous trial's stimulus $s_2(t-1)$, the larger is this attractive effect on average. Accordingly with panel (**A**), this effect is stronger for longer delay intervals (left to right panel). (**C**) The distribution of the bump displacement during delay period is characterized by two modes: a main one centered around 0, corresponding to correct trials where the working memory (WM) bump is not displaced during the delay interval, and another one centered around $s_1(t) - s_2(t-1)$, where the bump is displaced during WM (delay interval is randomly selected between 2, 4, and 10 s). We show here this distribution for three values of $s_2(t-1)$.

the drift present in the PPC network that eventually, for long enough delay intervals, causes the bump to arrive at the boundaries of the attractor, which would result in an error.

This simple analysis implies that contraction bias in the WM network in our model is not the result of the representation of the mean stimulus in the PPC, but is an effect that emerges as a result of the PPC network's sampling dynamics, mostly from recently presented stimuli. Indeed, a 'contraction to the mean' hypothesis only provides a general account of which pairs of stimuli should benefit from a better performance and which should suffer, but does not explain the gradual accumulation of errors upon increasing (decreasing) $s_1$, for pairs below (above) the $s_1 = s_2$ diagonal (**Fassihi et al., 2014**; **Fassihi et al., 2017**; **Akrami et al., 2018**). Notably, it cannot explain why the performance in trials with pairs of stimuli where $s_2$ is most distant from the mean stand to benefit the most from it. Altogether, our model suggests that contraction bias may be a simple consequence of errors occurring at single trials, driven by inputs from the PPC that follow a distribution similar to that of the external input (**Figure 4B**).

## Contraction bias in continuous recall

Can contraction bias also be observed in the activity of the WM network prior to binary decision-making? Many studies have evidenced contraction bias also in delayed estimation (or production) paradigms, where subjects must retain the value of a continuous parameter in WM and reproduce it after a delay (**Papadimitriou et al., 2015**; **Jazayeri and Shadlen, 2010**). Given that we observe contraction bias in the behavior of the network, we reasoned that this should also be evident prior to binary decision-making. Similar to delayed estimation tasks, we therefore analyzed the position of the bump $\hat{s}$, at the end of the delay interval, for each value of $s_1$. Consistent with our reasoning, we observe contraction bias of the value of $\hat{s}$, as evidenced by the systematic departure of the curve corresponding to the bump location from that of the nominal value of the stimulus (**Figure 5A**). We

also find that this contraction bias becomes greater as the delay interval increases (*Figure 5A*, right). We next analyzed the effect of the previous trial on the current trial by computing the displacement of the bump during the WM delay as a function of the distance between the current trial's stimulus and the previous trial's stimulus $s_1(t) - s_2(t-1)$ (*Figure 5B*). We found that when this distance is larger, the displacement of the bump during WM is on average also larger (*Figure 5B*). This displacement is also attractive. Breaking down these effects by delay, we find that longer delays lead to greater attraction (*Figure 5B*, right).

These results point to attractive effects of the previous trial, leading in turn to contraction bias in our model. To better understand the dynamics leading to them, we next looked at the distribution of bump displacements conditioned on a specific value of the second stimulus of the previous trial $s_2(t-1)$ (*Figure 5C*). These distributions are characterized by a mode around 0, corresponding to a majority of trials in which the bump is not displaced, and another mode around $s_1(t) - s_2(t-1)$, corresponding to the displacement in the direction of the preceding trial's stimulus described in section 'Multiple timescales at the core of short-term sensory history effects' and *Figure 2B*. However, note that the variance of this second mode can be large, reflecting displacements to locations other than $s_2(t-1)$, due to the complex dynamics in both networks that we have described in detail in section 'Multiple timescales at the core of short-term sensory history effects'.

## Model predictions

### The stimulus distribution impacts the pattern of contraction bias through its cumulative

In our model, the pattern of errors is determined by the cumulative distribution of stimuli from the correct decision boundary $s_1 = s_2$ to the left (right) for pairs of stimuli below (above) the diagonal (*Figure 4C* and *Figure 4—figure supplement 1A*). This implies that using a stimulus set in which this distribution is deformed makes different predictions for the gradient of performance across different stimulus pairs. A distribution that is symmetric (*Figure 4—figure supplement 1A*) yields an equal performance for pairs below and above the $s_1 = s_2$ diagonal (blue and red lines) when $s_1$ is at the mean (as well as the median, given the symmetry of the distribution). A distribution that is skewed, instead, yields an equal performance when $s_1$ is at the median for both pairs below and above the diagonal. For a negatively skewed distribution (*Figure 4—figure supplement 1B*) or positively skewed distribution (*Figure 4—figure supplement 1C*), the performance curves for pairs of stimuli below and above the diagonal show different concavity. For a distribution that is bimodal, the performance as a function of $s_1$ resembles a saddle, with equal performance for intermediate values of $s_1$ (*Figure 4—figure supplement 1D*). These results indicate that although the performance is quantitatively shaped by the form of the stimulus distribution, it persists as a monotonic function of $s_1$ under a wide variety of manipulations of the distributions. This is a result of the property of the cumulative function and may underlie the ubiquity of contraction bias under different experimental conditions.

We compare the predictions from our simple statistical model to the Bayesian model in *Loewenstein et al., 2021*, outlined in section 'Bayesian description of contraction bias'. We compute the predicted performance of an ideal Bayesian decision maker using a value of the uncertainty in the representation of the first stimulus ($\sigma = 0.12$) that yields the best fit with the performance of the statistical model (where the free parameter is $\epsilon = 0.5$; *Figure 4—figure supplement 1A–D*, panel 2). Our model makes different predictions across all types of distributions from that of the Bayesian model. Across all of the distributions (used as priors, in the Bayesian model), the main difference is that of a monotonic dependence of performance as a function of $s_1$ for our model (*Figure 4—figure supplement 1A–D*, panel 2). The biggest difference can be seen with a prior in which pairs of stimuli with extreme values are much more probable than middle-range values. Indeed, in the case of a bimodal prior, for pairs of stimuli where our model would predict a worse-than-average performance (*Figure 4—figure supplement 1D*, panel 3), the Bayesian model predicts a very good performance (*Figure 4—figure supplement 1D*, panel 4).

Do human subjects perform as predicted by our model (*Figure 6A*)? We tested 34 human subjects on the auditory modality of the task. The experimental protocol was identical to the one used in *Akrami et al., 2018*. Briefly, participants were presented with two sounds separated by a delay interval that varied across trials (randomly selected from 2, 4, and 6 s). After the second sound, participants were required to decide which sound was louder by pressing the appropriate key. We tested two groups of

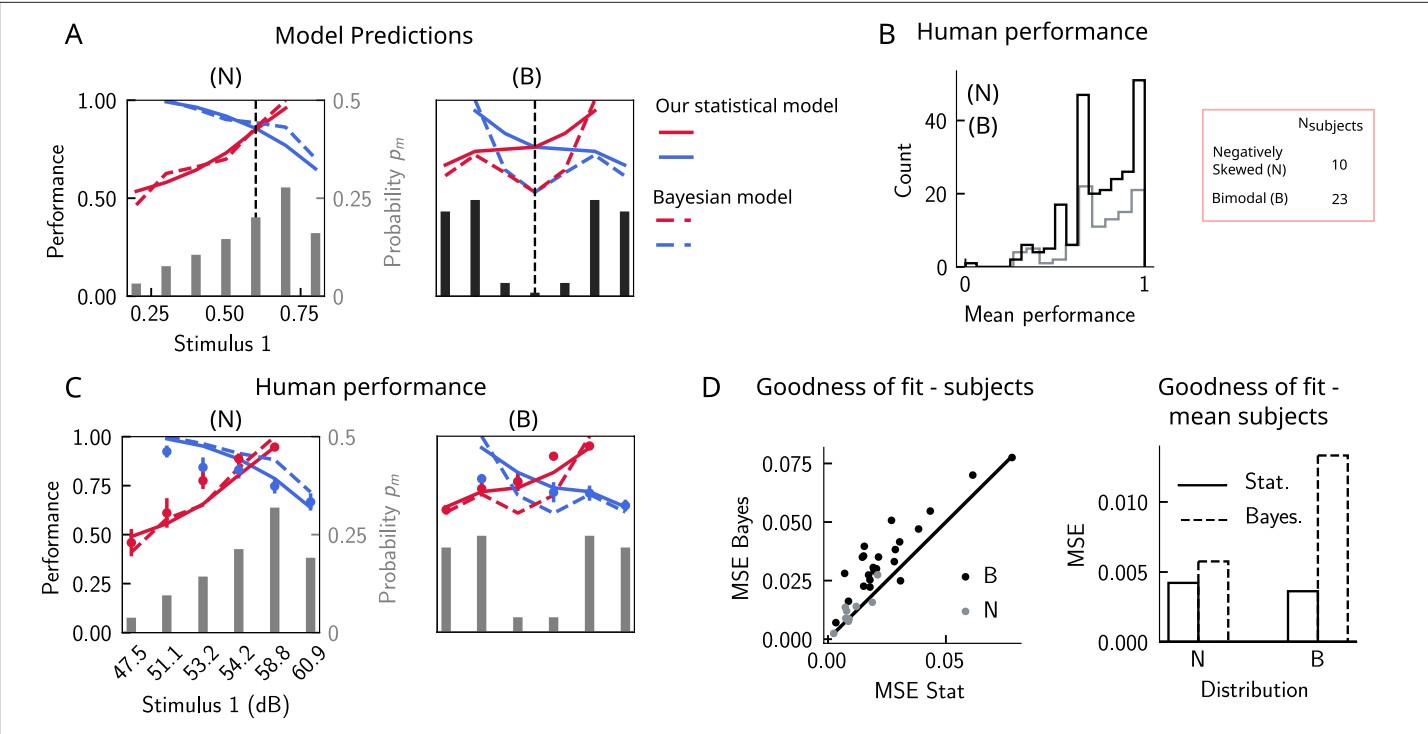

**Figure 6.** The stimulus distribution impacts the pattern of contraction bias through its cumulative. (**A**) Left panel: prediction of performance (left y-axis) of our statistical model (solid lines) and the Bayesian model (dashed lines) for a negatively skewed stimulus distribution (gray bars, to be read with the right y-axis). Blue (red): performance as a function of $s_1$ for pairs of stimuli where $s_1 > s_2$ ($s_1 < s_2$). Vertical dashed line: median of distribution. Right: same as left, but for a bimodal distribution. (**B**) The distribution of performance across different stimuli pairs and subjects for the negatively skewed (gray) and the bimodal distribution (black). On average, across both distributions, participants performed with an accuracy of 75%. (**C**) Left: mean performance of human subjects on the negatively skewed distribution (dots, error bars correspond to the standard deviation across different participants). Solid (dashed) lines correspond to fits of the mean performance of subjects with the statistical (Bayesian) model, $\epsilon = 0.55$ ($\sigma = 0.38$). Red (blue): performance as a function of $s_1$ for pairs of stimuli where $s_1 < s_2$ ($s_1 > s_2$), to be read with the left y-axis. The marginal stimulus distribution is shown in gray bars, to be read with the right y-axis. Right: same as left panel, but for the bimodal distribution. Here $\epsilon = 0.54$ ($\sigma = 0.73$). (**D**) Left: goodness of fit, as expressed by the mean-squared error (MSE) between the empirical curve and the fitted curve (statistical model in the x-axis and the Bayesian model in the y-axis) computed individually for each participant and each distribution. Right: goodness of fit computed for the average performance over participants in each distribution.

participants on two stimulus distributions: a negatively skewed and a bimodal distribution (*Figure 6A*, see section 'Human auditory experiment: delayed comparison task' for more details). Participants performed the task with a mean accuracy of approximately 75%, across stimulus distribution groups and across pairs of stimuli (*Figure 6B*). The experimental data was compatible with the predictions of our model. First, for the negatively skewed stimulus distribution condition, we observe a shift of the point of equal performance to the right, relative to a symmetric distribution (*Figure 6C*, left panel). For the bimodal condition, such a shift is not observed, as predicted by our model (*Figure 6C*, right panel). Second, the monotonic behavior of the performance, as a function of $s_1$, also holds across both distributions (*Figure 6C*). Our model provides a simple explanation: the percent correct on any given pair is given by the probability that, given a shift in the WM representation, this representation still does not affect the outcome of the trial (*Figure 4C*). This probability is given by cumulative of the probability distribution of WM representations for which we assume the marginal distribution of the stimuli to be a good approximation (*Figure 4A*). As a result, performance is a monotonic function of $s_1$, independent of the shape of the distribution, while the same does not always hold true for the Bayesian model (*Figure 6C*).

We further fit the performance of each participant using both our statistical model and the Bayesian model by minimizing the mean-squared error (MSE) loss between the empirical curve and the model, with $\epsilon$ and $\sigma$ as free parameters (*Figure 6C*), respectively (for the Bayesian model, we used the marginal distribution of the stimuli $p_m$ as the prior). Across participants in both distributions, our

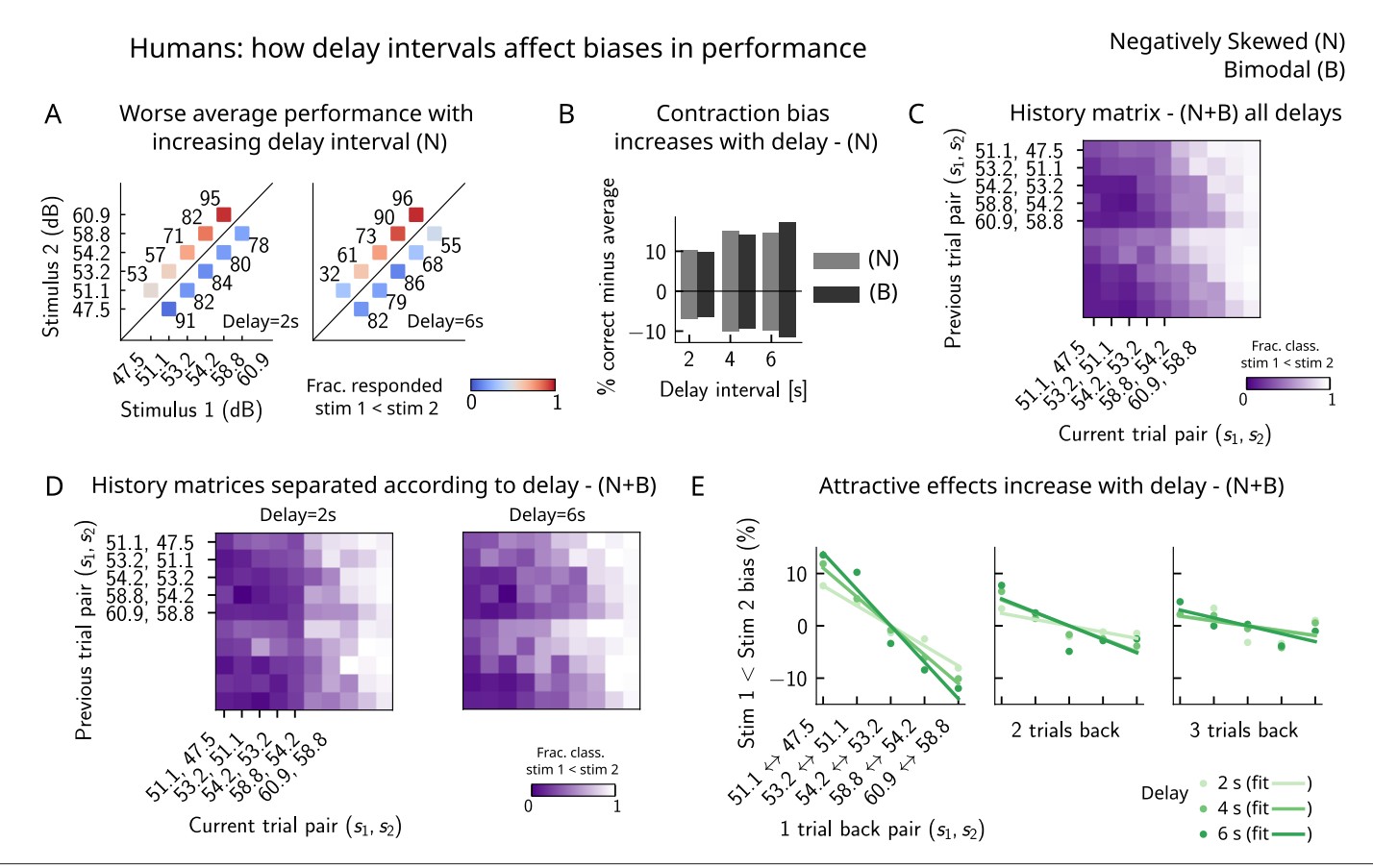

**Figure 7.** Attractive effects of the previous trials lead to contraction bias in human subjects, both increasing with delay interval. (**A**) The performance (in percentage correct, shown in numbers above each stimulus pair) of human subjects is better with lower delay intervals (left, 2 s) than with higher delay intervals (right, 6 s). Colorbar expresses the fraction of trials in which participants responded that $s_1 < s_2$. Results are for the negatively skewed stimulus distribution, noted (N). (**B**) Concurrently, contraction bias on Bias+ and Bias- trials (quantification explained in text) also increases with an increased delay interval for both stimulus distributions (negatively skewed in gray and bimodal in black). (**C**) History matrix expressing the fraction of trials in which subjects responded $s_1 < s_2$ (in color) for every pair of current (x-axis) and previous (y-axis) stimuli for negatively skewed and bimodal stimulus distributions (N+B). The one-trial back history effects can be seen through the vertical modulation of the color. Colorbar codes for the fraction of trials in which subjects responded $s_1 < s_2$. (**D**) History matrices (as in [**C**]) computed for all distributions and separated according to delay intervals (left: 2 s and right: 6 s). (**E**) Bias, quantifying the (attractive) effect of previous stimulus pairs, for 1–3 trials back in history. The attractive bias computed for all distributions increases with the delay interval separating the two stimuli (light to dark green: increasing delay).

statistical model yielded a better fit of the performance, relative to the Bayesian model (**Figure 6D**, left panel). We further fit the mean performance across all participants within a given distribution group and similarly found that the statistical model yields a better fit using the MSE as a goodness-of-fit metric (**Figure 6D**, right panel).

Finally, in order to better understand the parameters that affect the occurrence of errors in human participants, we computed the performance and fraction classified as $s_1 < s_2$ separately for different delay intervals. We found that the larger the delay interval, the lower the average performance (**Figure 7A**), accompanied by a larger contraction bias for larger intervals (**Figure 7B**). We further analyzed the fraction of trials in which subjects responded $s_1 < s_2$, conditioned on the specific pair of stimuli presented in the current and the previous trials (**Figure 7C**) for all distributions (one negatively skewed and two bimodal distributions, of which only one is shown in **Figure 6C**). Compatible with the previous results (**Akrami et al., 2018**), we found attractive history effects that increased with the delay interval (**Figure 7D and E**).

## A prolonged intertrial interval improves average performance and reduces attractive bias

If errors are due to the persistence of activity resulting from previous trials, what then is the effect of the ITI? In our model, a shorter ITI (relative to the default value of 6 s used in *Figures 2 and 3*) results in a worse performance and vice versa (*Figure 8A–C*). This change in performance is reflected in reduced biases toward the previous trial (*Figure 8D and E*). A prolonged ITI allows for a drifting bump to vanish due to the effect of adaptation: as a result, the performance improves with increasing ITI and conversely worsens with a shorter ITI.

Do human subjects express less bias with longer ITIs as predicted by our model? In our simulations, we set the ITI to either 2.2, 6, or 11 s, whereas in the experiment, since it is self-paced, the ITI can vary considerably. In order to emulate the simulation setting as closely as possible, we divided trials into two groups: 'short' ITIs (shorter than 3 s) and 'long' ITIs (longer than 3 s). This choice was motivated by the shape of the distribution of ITIs, which is bimodal, with a peak around 1 s, and another after 3 s (*Figure 8F*). Given the shape of the ITI distribution, we did not divide the ITIs into smaller intervals as this would result in too little data in some intervals. In line with our model, we found a better average performance with increasing ITI accompanied by decreasing contraction bias (*Figure 8G*). In order to quantify one-trial-back effects, we used data pertaining to all of the distributions we tested – the negatively skewed, and also two bimodal distributions (of which only one is shown in this article, in *Figure 6C*). This allowed us to obtain clear one-trial-back attractive biases, decreasing with increasing ITI (*Figure 8H*), in line with our model predictions (*Figure 8B and D*).

## Working memory is attracted toward short-term and repelled from long-term sensory history

Although contraction bias is robustly found in different contexts, surprisingly similar tasks, such as perceptual estimation tasks, sometimes highlight opposite effects, that is, repulsive effects (*Fritsche et al., 2017*; *Li et al., 2017*; *Hachen et al., 2021*). Interestingly, recent studies have found both effects in the same experiment: in a study of visual orientation estimation (*Fritsche and Spaak, 2020*), it has been found that attraction and repulsion have different timescales; while perceptual decisions about orientation are attracted toward recently perceived stimuli (timescale of a few seconds), they are repelled from stimuli that are shown further back in time (timescale of a few minutes). Moreover, in the same study, they find that the long-term repulsive bias is spatially specific, in line with sensory adaptation (*Knapen et al., 2010*; *Boi et al., 2011*; *Mathôt and Theeuwes, 2013*) and in contrast to short-term attractive serial dependence (*Fritsche and Spaak, 2020*). Given that adaptation is a main feature of our model of the PPC, we sought to determine whether such repulsive effects can emerge from the model. We extended the calculation of the bias to up to 10 trials back and quantified the slope of the bias as a function of the previous trial stimulus pair. We observe robust repulsive effects appear after the third trial back in history and up to six trials back (*Figure 8I*). In our model, both short-term attractive effects and longer-term repulsive effects can be attributed to the multiple timescales over which the networks operate. The short-term attractive effects occur due to the long time it takes for the adaptive threshold to build up in the PPC and the short timescale with which the WM network integrates input from the PPC. The longer-term repulsive effects occur when the activity bump in the PPC persists in one location and causes adaptation to slowly build up, effectively increasing the activation threshold. The raised threshold takes equally long to return to baseline, preventing activity bumps to form in that location and thereby creating repulsion toward all the other locations in the network. Crucially, however, the amplitude of such effects depends on the ITI; in particular, for shorter ITIs, the repulsive effects are less observable.

### The timescale of adaptation in the PPC network can control perceptual biases similar to those observed in dyslexia and autism

In a recent study (*Lieder et al., 2019*), a similar PWM task with auditory stimuli was studied in human neurotypical (NT), autistic spectrum (ASD) and dyslexic (DYS) subjects. Based on an analysis using a generalized linear model (GLM), a double dissociation between different subject groups was suggested: ASD subjects exhibit a stronger bias toward long-term statistics – compared to NT subjects – while for DYS subjects, a higher bias is present toward short-term statistics.

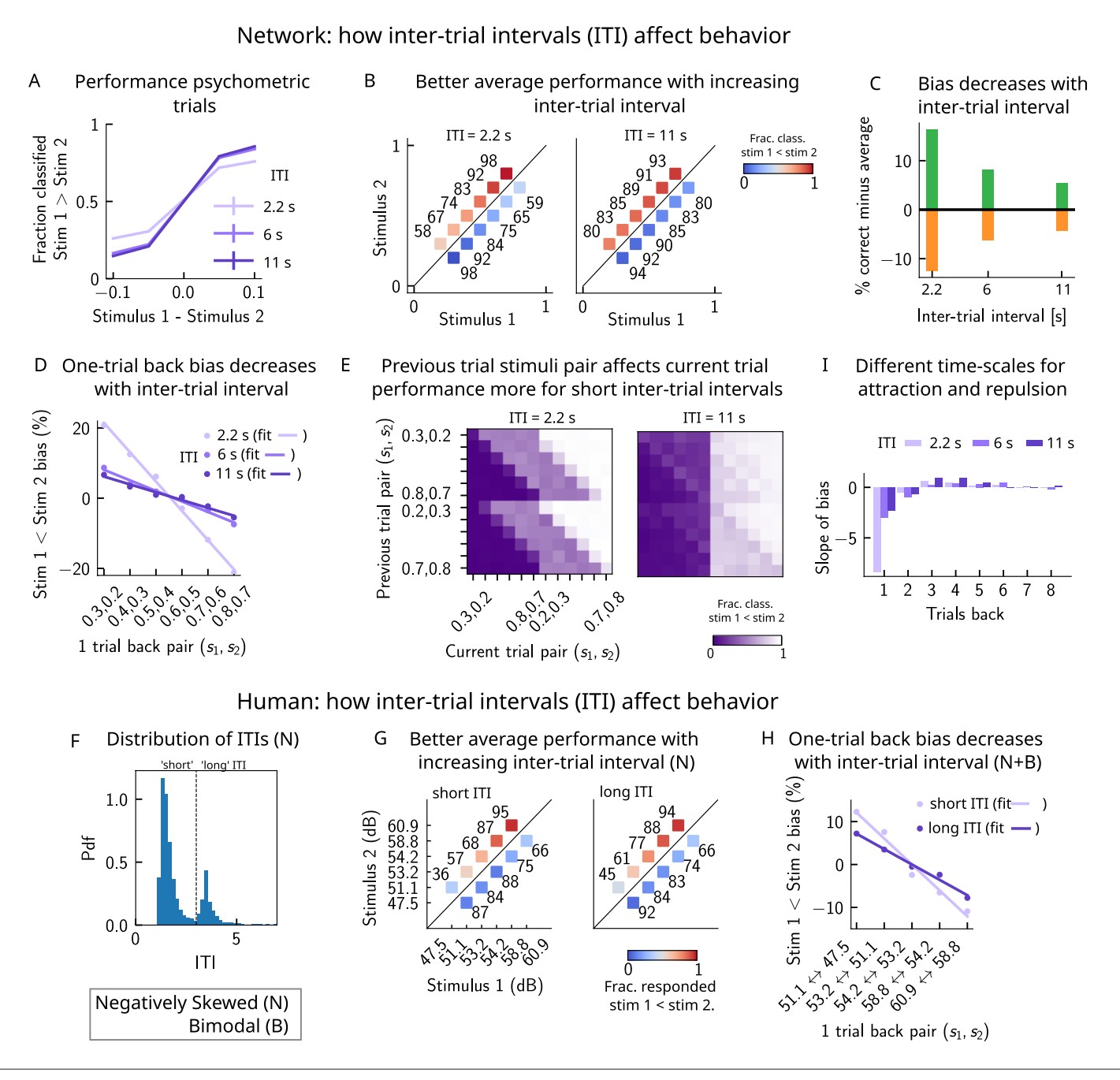

**Figure 8.** A prolonged intertrial interval (ITI) improves average performance and reduces attractive biases. Working memory is attracted toward short-term and repelled from long-term sensory history. (**A**) Performance of the network model for the psychometric stimuli improves with an increasing ITI. Error bars (not visible) correspond to the SEM over different simulations. (**B**) The network performance (numbers next to stimulus pairs) is on average better for longer ITIs (right panel, ITI = 11 s) compared to shorter ones (left panel, ITI = 2.2 s). Colorbar indicates the fraction of trials classified as $s_1 < s_2$. (**C**) Quantifying contraction bias separately for Bias+ trials (green) and Bias- trials (orange) yields a decreasing bias as the ITI increases. (**D**) The bias, quantifying the (attractive) effect of the previous trial, decreases with ITI. Darker shades of purple correspond to increasing values of the ITI, with dots corresponding to simulation values and lines to linear fits. (**E**) Performance is modulated by the previous stimulus pairs (modulation along the y-axis), more for a short ITI (left, ITI = 2.2 s) than for a longer ITI (right, IT I = 11 s). The colorbar corresponds to the fraction classified $s_1 < s_2$. (**F**) The distribution of ITIs in the human experiment is bimodal. We define as having a 'short' ITI, those trials where the preceding ITI is shorter than 3 s and conversely for 'long' ITI. (**G**) The human performance for the negatively skewed stimulus distribution is on average worse for shorter ITIs (left panel) compared to longer ones (right panel). Colorbar indicates the fraction of trials subjects responded $s_1 < s_2$. (**H**) The bias, quantifying the (attractive) effect of the previous trial, increases with ITI in human subjects. Darker shades of purple correspond to increasing values of the ITI, with dots corresponding

*Figure 8 continued on next page*

*Figure 8 continued*

to empirical values and lines to linear fits. (**I**) Although the stimuli shown up to two trials back yield attractive effects, those further back in history yield repulsive effects, notably when the ITI is larger. Such repulsive effects extend to up to six trials back.

We investigated our model to see whether it is able to show similar phenomenology, and if so, what are the relevant parameters controlling the timescale of the biases in behavior? We identified the adaptation timescale in the PPC as the parameter that affects the extent of the short-term bias, consistent with previous literature (*Jaffe-Dax et al., 2018*; *Jaffe-Dax et al., 2017*). Calculating the mean bias toward the previous trial stimulus pair (*Figure 9A*), we find that a shorter-than-NT adaptation timescale yields a larger bias toward the previous trial stimulus. Indeed, a shorter timescale for neuronal adaptation implies a faster process for the extinction of the bump in PPC – and the formation of a new bump that remains stable for a few trials – producing 'jumpier' dynamics that lead to a larger number of one-trial-back errors. In contrast, increasing this timescale with respect to NT gives rise to a stable bump for a longer time, ultimately yielding a smaller short-term bias. This can be seen in the detailed breakdown of the network's behavior on the current trial when conditioned on the stimuli presented at the previous trial (*Figure 9B*, see also section 'Multiple timescales at the core of

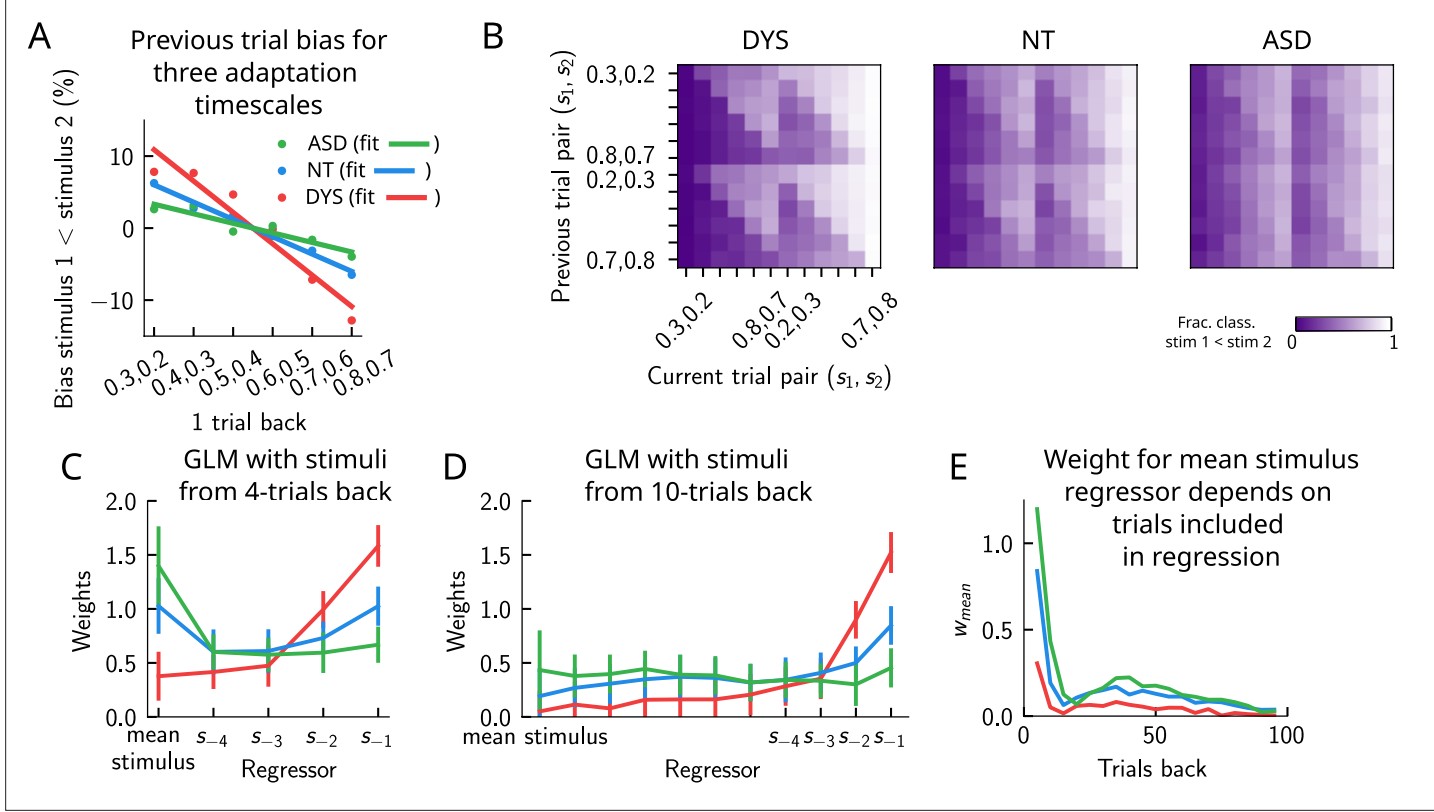

**Figure 9.** Apparent tradeoff between short- and long-term biases controlled by the timescale of neural adaptation. (**A**) The bias exerted on the current trial by the previous trial (see main text for how it is computed) for three values of the adaptation timescale that mimic similar behavior to the three cohorts of subjects. (**B**) As in *Figure 2D*, for three different values of adaptation timescale. The colorbar corresponds to the fraction classified $s_1 < s_2$. (**C**) Generalized linear model (GLM) weights corresponding to the three values of the adaptation parameter marked in *Figure 9—figure supplement 1A*, including up to four trials back. In a GLM variant incorporating a small number of past trials as regressors, the model yields a high weight for the running mean stimulus regressor. Error bars correspond to the standard deviation across different simulations. (**D**) Same as in (**C**), but including regressors corresponding to the past 10 trials as well as the running mean stimulus. With a larger number of regressors extending into the past, the model yields a small weight for the running mean stimulus regressor. Error bars correspond to the standard deviation across different simulations. (**E**) The weight of the running mean stimulus regressor as a function of extending the number of past trial regressors decays upon increasing the number of previous trial stimulus regressors.

The online version of this article includes the following figure supplement(s) for figure 9:

**Figure supplement 1.** Apparent tradeoff between short- and long-term biases controlled by the timescale of neural adaptation.

short-term sensory history effects' for a more detailed explanation of the dynamics). We performed a GLM analysis as in *Lieder et al., 2019* to the network behavior, with stimuli from four trials back and the mean stimulus as regressors (see section 'Generalized linear model'). This analysis shows that a reduction in the PPC adaptation timescale with respect to NT produces behavioral changes qualitatively compatible with data from DYS subjects; on the contrary, an increase of this timescale yields results consistent with ASD data (*Figure 9C*).

This GLM analysis suggests that dissociable short- and long-term biases may be present in the network behavior. Having access to the full dynamics of the network, we sought to determine how it translates into such dissociable short- and long-term biases. Given that all the behavior arises from the location of the bump on the attractor, we quantified the fraction of trials in which the bump in the WM network, before the onset of the second stimulus, was present in the vicinity of any of the previous trial's stimuli (*Figure 9—figure supplement 1B*, right panel, and *Figure 9—figure supplement 1C*), as well as the vicinity of the mean over the sensory history (*Figure 9—figure supplement 1B*, left panel, and *Figure 9—figure supplement 1C*). While the bump location correlated well with the GLM weights corresponding to the previous trial's stimuli regressor (comparing the right panels of *Figure 9—figure supplement 1A and B*), surprisingly, it did not correlate with the GLM weights corresponding to the mean stimulus regressor (comparing the left panels of *Figure 9—figure supplement 1A and B*). In fact, we found that the bump was in a location given by the stimuli of the past two trials, as well as the mean over the stimulus history, in a smaller fraction of trials, as the adaptation timescale parameter was made larger (*Figure 9—figure supplement 1C*).

Given that the weights, after four trials in the past, were still non-zero, we extended the GLM regression by including a larger number of past stimuli as regressors. We found that doing this greatly reduced the weight of the mean stimulus regressor (*Figure 9C–E*, see section 'Generalized linear mode' for more details). Therefore, we propose an alternative interpretation of the GLM results given in *Lieder et al., 2019*. In our model, the increased (reduced) weight for long-term mean in the ASD (DYS) subjects can be explained as an effect of a larger (smaller) window in time of short-term biases without invoking a double dissociation mechanism (*Figure 9D and E*). In section 'Generalized linear model', we provide a mathematical argument for this, which is empirically shown by including a large number of individual stimuli from previous trials in the regression analysis.

## Discussion

### Contraction bias in the delayed comparison task: simply a statistical effect or more?

Contraction bias is an effect emerging in WM tasks, where in the averaged behavior of a subject the magnitude of the item held in memory appears to be larger than it actually is when it is 'small' and, vice versa, it appears to be smaller when it is 'large' (*Algom, 1992*; *Berliner et al., 1977*; *Hellström, 1985*; *Poulton and Poulton, 1989*; *Ashourian and Loewenstein, 2011*; *Preuschhof et al., 2010*; *Olkkonen et al., 2014*). Recently, *Akrami et al., 2018* found that contraction bias as well as short-term history-dependent effects occur in an auditory delayed comparison task in rats and humans: the comparison performance in a given trial depends on the stimuli shown in preceding trials (up to three trials back) (*Akrami et al., 2018*), similar to previous findings in human 2AFC paradigms (*Raviv et al., 2012*). These findings raise the question: does contraction bias occur independently of short-term history effects, or does it emerge as a result of the latter?

*Akrami et al., 2018* have also found the PPC to be a critical node for the generation of such effects as its optogenetic inactivation (specifically during the delay interval) greatly attenuated both effects. WM was found to remain intact, suggesting that its content was perhaps read-out in another region. Electrophysiological recordings as well as optogenetic inactivation results in the same study suggest that while sensory history information is provided by the PPC, its integration with the WM content must happen somewhere downstream to the PPC. Different brain areas can fit the profile. For instance, there are known projections from the PPC to mPFC in rats (*Olsen et al., 2019*), where neural correlates of PWM have been found (*Esmaeili and Diamond, 2019*). Building on these findings, we suggest a minimal two-module model aimed at better understanding the interaction between contraction bias and short-term history effects. These two modules capture properties of the PPC (in providing sensory history signals) and a downstream network holding WM content. Our WM and

PPC networks, despite having different timescales, are both shown to encode information about the marginal distribution of the stimuli (*Figure 4A*). Although they have similar activity distributions to that of the external stimuli, they have different memory properties due to the different timescales with which they process incoming stimuli. The putative WM network, from which information to solve the task is read-out, receives additional input from the PPC network. The PPC is modeled as integrating inputs slower relative to the WM network and is also endowed with firing rate adaptation, the dynamics of which yield short-term history biases and, consequently, contraction bias.

It must be noted, however, that short-term history effects (due to firing rate adaptation) do not necessarily need to be invoked in order to recover contraction bias: as long as errors are made following random samples from a distribution in the same range as that of the stimuli, contraction bias should be observed (*Tong and Dubé, 2022*). Indeed, when we manipulated the parameters of the PPC network in such a way that short-term history effects were eliminated (by removing the firing rate adaptation), contraction bias persisted. As a result, our model suggests that contraction bias may not simply be given by a regression toward the mean of the stimuli during the interstimulus interval (*Karim et al., 2013*; *Kerst and Howard, 1978*), but brought about by a richer dynamics occurring at the level of individual trials (*Jou et al., 2004*), more in line with the idea of random sampling (*Rahnev and Denison, 2018*).

The model makes predictions as to how the pattern of errors may change when the distribution of stimuli is manipulated either at the level of the presented stimuli or through the network dynamics. When we tested these predictions experimentally by manipulating the skewness of the stimulus distribution such that the median and the mean were dissociated (*Figure 6A*), the results from our human psychophysics experiments were in agreement with the model predictions. In further support of this, in a recent tactile categorization study (*Hachen et al., 2021*), where rats were trained to categorize tactile stimuli according to a boundary set by the experimenter, the authors have shown that rats set their decision boundary according to the statistical structure of the stimulus set to which they are exposed. More studies are needed to fully verify the extent to which the statistical structure of the stimuli affects the performance. Finally, we note that in our model the stimulus distribution is not explicitly learned (but see *Maes et al., 2023*): instead, the PPC dynamics follows the input, and its marginal distribution of activity is similar to that of the external input. This is in agreement with *Hachen et al., 2021*, where the authors used different stimulus ranges across different sessions and noted that rats initiated each session without any residual influence of the previous session's range/ boundary on the current session, ruling out long-term learning of the input structure.

Importantly, our results are not limited to the delayed 'comparison' paradigm, where binary decision-making occurs. We show that by analyzing the location of the WM bump at the end of the delay interval, similar to the continuous recall tasks, we can retrieve the averaged effects of contraction bias, similar to previous reports (*Jazayeri and Shadlen, 2010*). Such continuous read-out of the memory reveals a rich dynamics of errors at the level of individual trials, similar to the delayed comparison case, but to our knowledge this has not been studied in previous experimental studies. *Papadimitriou et al., 2015* have characterized residual error distribution in an orientation recall task when limiting previous trials to orientations in the range of +35 to+ 85° relative to the current trial. This distribution is unimodal, leading the authors to conclude that the current trial shows a small but systematic bias toward the location of the memorandum of the previous trial. It remains to be tested whether the error distribution remains unimodal if conditioned on other values of the current and previous orientations, similar to our analysis in *Figure 5C*.

## Attractor mechanism riding on multiple timescales

Our model assumes that the stimulus is held in WM through the persistent activity of neurons, building on the discovery of persistent selective activity in a number of cortical areas, including the prefrontal cortex (PFC), during the delay interval (*Fuster and Alexander, 1971*; *Miyashita and Chang, 1988*; *Funahashi et al., 1989*; *Funahashi et al., 1990*; *Romo et al., 1999*; *Salinas et al., 2000*; *Zhang et al., 2019*). To explain this finding, we have used the attractor framework, in which recurrently connected neurons mutually excite one another to form reverberation of activity within populations of neurons coding for a given stimulus (*Hopfield, 1982*; *Amit, 1992*; *Battaglia and Treves, 1998*). However, subsequent work has shown that persistent activity related to the stimulus is not always present during the delay period and that the activity of neurons displays far more heterogeneity than previously

thought (*Barak et al., 2013*). It has been proposed that short-term synaptic facilitation may dynamically operate to bring a WM network across a phase transition from a silent to a persistently active state (*Mongillo et al., 2008*; *Barak and Tsodyks, 2007*). Such mechanisms may further contribute to short-term biases (*Barbosa et al., 2020*), an alternative possibility that we have not specifically considered in this model.

An important model feature that is crucial in giving rise to all of its behavioral effects is its operation over multiple timescales (*Figure 3—figure supplement 2F*). Such timescales have been found to govern the processing of information in different areas of the cortex (*Murray et al., 2014*; *Siegle et al., 2021*; *Gao et al., 2020*) and may reflect the heterogeneity of connections across different cortical areas (*Stern et al., 2021*).

## Relation to other models

In many early studies, groups of neurons whose activity correlates monotonically with the stimulus feature, known as 'plus' and 'minus' neurons, have been found in the PFC (*Romo et al., 1999*; *Barak et al., 2010*). Such neurons have been used as the starting point in the construction of many models (*Miller et al., 2003*; *Machens et al., 2005*; *Barak et al., 2013*; *Barak and Tsodyks, 2014*). It is important, however, to note that depending on the area the fraction of such neurons can be small (*Esmaeili and Diamond, 2019*) and that the majority of neurons exhibit firing profiles that vary largely during the delay period (*Machens et al., 2010*). Such heterogeneity of the PFC neurons' temporal firing profiles has prompted the successful construction of models that have not included the basic assumption of plus and minus neurons, but these have largely focused on the plausibility of the dynamics of neurons observed, with little connection to behavior (*Barak et al., 2013*).

A separate line of research has addressed behavior by focusing on normative models to account for contraction bias (*Ashourian and Loewenstein, 2011*; *Raviv et al., 2012*; *Rahnev and Denison, 2018*; *Salinas, 2011*). The abstract mathematical model that we present (*Figure 4*) can be compatible with a Bayesian framework (*Ashourian and Loewenstein, 2011*) in the limit of a very broad likelihood for the first stimulus and a very narrow one for the second stimulus, and where the prior for the first stimulus is replaced by the distribution of $\hat{s}$, following the model in *Figure 4B* (see section 'The probability to make errors is proportional to the cumulative distribution of the stimuli, giving rise to contraction bias' for details). However, it is important to note that our model is conceptually different, that is, subjects do not have access to the full prior distribution, but only to samples of the prior. We show that having full knowledge of the underlying sensory distribution is not needed to present contraction bias effects. Instead, a point estimate of past events that is updated trial to trial suffices to show similar results. This suggests a possible mechanism for the brain to approximate Bayesian inference, and it remains open whether similar mechanisms (based on the interaction of networks with different integration timescales) can approximate other Bayesian computations. It is also important to note the differences between the predictions from the two models. As shown in *Figure 6A* and *Figure 4—figure supplement 1*, depending on the specific sensory distributions, the two models can have qualitatively different testable predictions. Data from our human psychophysical experiments, utilizing auditory PWM, show better agreement with our model predictions compared to the Bayesian model.

Moreover, an ideal Bayesian observer model alone cannot capture the temporal pattern of short-term attraction and long-term repulsion observed in some tasks, and the model has had to be supplemented with efficient encoding and Bayesian decoding of information in order to capture both effects (*Fritsche and Spaak, 2020*). In our model, both effects emerge naturally as a result of neuronal adaptation, but their amplitudes crucially depend on the time parameters of the task, perhaps explaining the sometimes contradictory effects reported across different tasks.

Finally, while such attractive and repulsive effects in performance may be suboptimal in the context of a task designed in a laboratory setting, this may not be the case in more natural environments. For example, it has been suggested that integrating information over time serves to preserve perceptual continuity in the presence of noisy and discontinuous inputs (*Fischer and Whitney, 2014*). This continuity of perception may be necessary to solve more complex tasks or make decisions, particularly in a non-stationary environment, or in a noisy environment.

## Methods

### The model

Our model is composed of two populations of $N$ neurons, representing the PPC network and the putative WM network. We consider that each population is organized as a continuous line attractor, with recurrent connectivity described by an interaction matrix $J_{ij}$, whose entries represent the strength of the interaction between neurons $i$ and $j$. The activation function of the neurons is a logistic function, that is, the output $r_i$ of neuron $i$, given the input $h_i$, is

$$r_i = \frac{1}{1 + e^{-\beta h_i}} \tag{1}$$

where $\beta$ is the neuronal gain. The variables $r_i$ take continuous values between 0 and 1 and represent the firing rates of the neurons. The input $h_i$ to a neuron is given by

$$\tau \frac{dh_i}{dt} + h_i = \sum_{j(\neq i)} J_{ij} r_j + I_i^{\text{ext}} \tag{2}$$

where $\tau$ is the timescale for the integration of inputs. In the first term on the right-hand side, $J_{ij} r_j$ represents the input to neuron $i$ from neuron $j$, and $I_i^{\text{ext}}$ corresponds to the external inputs. The recurrent connections are given by

$$J_{ij} = \frac{1}{d_0}(K_{ij} - J_0), \tag{3}$$

with

$$K_{ij} = J_e \, e^{-\frac{|x_i - x_j|}{d_0}}. \tag{4}$$

The interaction kernel, $K$, is assumed to be the result of a time-averaged Hebbian plasticity rule: neurons with nearby firing fields will fire concurrently and strengthen their connections, while firing fields far apart will produce weak interactions (*Dalgleish et al., 2020*). Neuron $i$ is associated with the firing field $x_i = i/N$. The form of $K$ expresses a connectivity between neurons $i$ and $j$ that is exponentially decreasing with the distance between their respective firing fields, proportional to $|i - j|$; the exponential rate of decrease is set by the constant $d_0$, that is, the typical range of interaction. The amplitude of the kernel is also rescaled by $d_0$ in such a way that $\sum_{i,j} K_{ij}$ is constant. The strength of the excitatory weights is set by $J_e$; the normalization of $K$, together with the sigmoid activation function saturating to 1, implies that $J_e$ is also the maximum possible input received by any neuron due to the recurrent connections. The constant $J_0$, instead, contributes to a linear global inhibition term. Its value needs to be chosen depending on $J_e$ and $d_0$, so that the balance between excitatory and inhibitory inputs ensures that the activity remains localized along the attractor, that is, it does not either vanish or equal 1 everywhere; together, these three constants set the width of the bump of activity.

The two networks in our model are coupled through excitatory connections from the PPC to the WM network. Therefore, we introduce two equations analogous to *Equation 2*, one for each network. The coupling between the two will enter as a firing rate-dependent input, in addition to $I^{\text{ext}}$. The dynamics of the input to a neuron in the WM network writes

$$\tau_h^W \frac{dh_i^W}{dt} + h_i^W = \sum_{j \in j^W} J_{ij} r_j^W + J^{P \rightarrow W} r_i^P + I_i^{\text{ext}}, \tag{5}$$

where $j^W$ indexes neurons in the WM network, and $\tau_h^W$ is the timescale for the integration of inputs in the WM network. The first term in the right-hand side corresponds to inputs from recurrent connections within the WM network. The second term corresponds to inputs from the PPC network. Finally, the last term corresponds to the external inputs used to give stimuli to the network. Similarly, for the PPC network we have

$$\tau_h^P \frac{dh_i^P}{dt} + h_i^P = \sum_{j \in j^P} J_{ij} r_j^P - \theta_i^P + I_i^{\text{ext}},$$ (6)

where $j^P$ indexes neurons in the PPC, and $\tau_h^P$ is the timescale for the integration of inputs in the PPC network; importantly, we set this to be longer than the analogous quantity for the WM network, $\tau_h^W < \tau_h^P$ (see **Appendix 1—table 1**). The first and third terms in the right-hand side are analogous to the corresponding ones for the WM network: inputs from within the network and from the stimuli. The second term instead corresponds to adaptive thresholds with dynamics specified by

$$\tau_\theta^P \frac{d\theta_i^P}{dt} + \theta_i^P = D^P r_i^P$$ (7)

modeling neuronal adaptation, where $\tau_\theta^P$ and $D^P$ set its timescale and its amplitude. We are interested in the condition where the timescale of the evolution of the input current is much smaller relative to that of the adaptation ($\tau_h^P \ll \tau_\theta^P$). For a constant $\tau_\theta^P$, we find that depending on the value of $D^P$, the bump of activity shows different behaviors. For low values of $D^P$, the bump remains relatively stable (**Figure 3—figure supplement 1C**; **Hollingworth, 1910**). Upon increasing $D^P$, the bump gradually starts to drift (**Figure 3—figure supplement 1C**; **Jou et al., 2004**; **Berliner et al., 1977**; **Romani and Tsodyks, 2015**). Upon increasing $D^P$ even further, a phase transition leads to an abrupt dissipation of the bump (**Figure 3—figure supplement 1C**; **Hellström, 1985**).

Note that, while the transition from bump stability to drift occurs gradually, the transition from drift to dissipation is abrupt. This abruptness in the transition from the drift to the dissipation regime may imply that only one of the two behaviors is possible in our model of the PPC (section 'Multiple timescales at the core of short-term sensory history effects'). In fact, our network model of the PPC operates in the 'drift' regime ($\tau_\theta^P = 7.5$, $D^P = 0.3$). However, we also observe dissipation of the bump, which is mainly responsible for the jumps observed in the model. This occurs due to the inputs from incoming external stimuli that affect the bump via the global inhibition in the model (**Figure 3—figure supplement 1A**). Therefore, external stimuli can allow the network to temporarily cross the sharp drift/dissipation boundary shown in **Figure 3—figure supplement 1B**. As a result, the combined effects of adaptation, together with external inputs and global inhibition, result in the drift/jump dynamics described in the main text.

Finally, both networks have a linear geometry with free boundary conditions, that is, no condition is imposed on the profile activity at neuron 1 or $N$.

## Simulation

We performed all the simulations using custom Python code. Differential equations were numerically integrated with a time step of $dt = 0.001$ using the forward Euler method. The activity of neurons in both circuits was initialized to $r = 0$. Each stimulus was presented for 400 ms. A stimulus is introduced as a 'box' of unit amplitude and of width $2\delta s$ around $s$ in stimulus space: in a network with $N$ neurons, the stimulus is given by setting $I_i^{\text{ext}} = 1$ in **Equation 5** for neurons with index $i$ within $(s \pm \delta s) \times N$, and $I_i^{\text{ext}} = 0$ for all the others. Only the activity in the WM network was used to assess performance. To do that, the activity vector was recorded at two timepoints: 200 ms before and after the onset of the second stimulus $s_2$. Then, the neurons with the maximal activity were identified at both timepoints and compared to make a decision. This procedure was done for 50 different simulations with 1000 consecutive trials in each, with a fixed ITI separating two consecutive trials, fixed to 5 s. The interstimulus intervals were set according to two different experimental designs, as explained below.

### Interleaved design

As in the study in **Akrami et al., 2018**, an interstimulus interval of either 2, 6, or 10 s was randomly selected. The delay interval is defined as the time elapsed from the end of the first stimulus to the beginning of the second stimulus. This procedure was used to produce **Figures 1–3 and 7**, **Figure 2—figure supplement 1**, and **Figure 3—figure supplement 2**.

## Block design

In order to provide a comparison to the interleaved design, but also to simulate the design in *Lieder et al., 2019*, we also ran simulations with a block design, where the interstimulus intervals were kept fixed throughout the trials. Other than this, the procedure and parameters used were exactly the same as in the interleaved case. This procedure was used to produce *Figure 9*, *Figure 2—figure supplement 2*, and *Figure 9—figure supplement 1*.

## Human auditory experiment: Delayed comparison task

Subjects received, in each trial, a pair of sounds played from ear-surrounding headphones. The subject self-initiated each trial by pressing the spacebar on the keyboard. The first sound was then presented together with a blue square on the left side of a computer monitor in front of the subject. This was followed by a delay period, indicated by 'WAIT!' on the screen, then the second sound was presented together with a red square on the right side of the screen. At the end of the second stimulus, subjects had 2 s to decide which one was louder, then indicate their choice by pressing the 's' key if they thought that the first sound was louder, or the 'l' key if they thought that the second sound was louder. Written feedback about the correctness of their response was provided on the screen for each individual trial. Every 10 trials, participants received feedback on their running mean performance calculated up to that trial. Participants then had to press spacebar to go to the next trial (the experiment was hence self-paced).

The two auditory stimuli, $s_1$ and $s_2$, separated by a variable delay (of 2, 4, and 6 s), were played for 400 ms, with short delay periods of 250 ms inserted before $s_1$ and after $s_2$. The stimuli consisted of broadband noise 2000–20,000 Hz, generated as a series of sound pressure level (SPL) values sampled from a zero-mean normal distribution. The overall mean intensity of sounds varied from 60 to 92 dB. Participants had to judge which out of the two stimuli, $s_1$ and $s_2$, was louder (had the greater SPL standard deviation). We recruited 10 subjects for the negatively skewed distribution and 24 subjects for the bimodal distribution. Each participant performed approximately 400 trials for a given distribution. Several participants took part in both distributions.

The study was approved by the University College London (UCL) Research Ethics Committee (16159/001) (London, UK). Before starting the experiment, participants were provided with an information sheet relevant to the experiment they will be performing and asked to sign an informed written consent. By signing this consent form, the participants consent to freely take part in the study and confirm they understand the information they received. The participants additionally confirm they understand that their participation is voluntary and that they are allowed to stop the experiment at any moment and withdraw the data they provided. Participants consent to allow use of their personal data only for the purpose of scientific research, as determined by applicable law. Identifiable personal data are only available to the researchers and securely stored upon need, while the anonymized version of this data is freely available in a public repository. Once published, the participants' contribution will remain non-identifiable.

## Computing bump location

In order to check whether the bump is in a target location (*Figure 3B*, *Figure 2—figure supplement 1D*, and *Figure 3—figure supplement 2B*), we check whether the position of the neuron with the maximal firing rate is within a distance of ±5% of the length of the whole line attractor from the target location (*Figure 3A*, *Figure 2—figure supplement 1C*, and *Figure 3—figure supplement 2A*). In these figures, we compare the probability that, in a given trial, the activity of the WM network is localized around one of the previous stimuli (estimated from the simulation of the dynamics, histograms) with the probability of this happening due to chance (horizontal dashed line). Here we detail the calculation of the chance probability. In general, if we have two discrete independent random variables, $\hat{X}$ and $\hat{Y}$, with probability distributions $p_X$ and $p_Y$, the probability of them having the same value is

$$\text{Prob}\{\hat{X} = \hat{Y}\} = \sum_{i,j} \underbrace{\text{Prob}\{\hat{X} = x_i\}}_{p_X^i} \underbrace{\text{Prob}\{\hat{Y} = y_j\}}_{p_Y^j} \mathbb{I}(x_i = y_j)$$

where $i, j$ are the indices for different values of the two random variables and $\mathbb{I}(x_i = y_j)$ equals 1 where $x_i = x_j$ and 0 otherwise. If the two random variables are identically distributed, the above expression writes

$$\text{Prob}\{\hat{X} = \hat{Y}\} = \sum_{i,j} p_X^i p_Y^j \, \delta_{i,j} = \sum_i \left(p_X^i\right)^2$$

In our case, the two identically distributed random variables are 'bump location at the current trial' and the 'target bump location' (that are $s_1^{t-2}$, $s_2^{t-2}$, $s_1^{t-1}$, $s_2^{t-1}$, and $\langle s \rangle$). With the exception of the mean stimulus $\langle s \rangle$, all the other variables are identically distributed, with probability $p_m$ (that is the marginal distribution over $s_1$ or $s_2$). We note that the bump location in the WM network follows a very similar distribution to $p_m$ (*Figure 4A*). Then, we compute the chance probability with the above relationship, where $p_X \equiv p_m$. For the mean stimulus, instead, we have a probability which is simply equal to 1 for $s = 0.5$ and 0 elsewhere; therefore, the chance probability for the bump location to be at the mean stimulus then is $p_m(0.5)$.

The excess probability (with respect to chance) for the bump location to equal one of the previous stimuli gives a measure of the correlation between these two; in other terms, of the amount of information retained by the network about previous stimuli.

## The probability to make errors is proportional to the cumulative distribution of the stimuli, giving rise to contraction bias

In order to illustrate the statistical origin of contraction bias consistent with our network model, we consider a simplified mathematical model of its performance (*Figure 4B*). By definition of the delayed comparison task, the optimal decision maker produces a label $y$ equal to 1 if $s_1^t < s_2^t$, and 0 if $s_1^t > s_2^t$; the impossible cases $s_1^t = s_2^t$ are excluded from the set of stimuli, but would produce a label which is either 0 or 1 with 50% probability. That is,

$$y(s_1, s_2) = \begin{cases} 1 & \text{if } s_1 < s_2 \\ 0 & \text{if } s_1 > s_2 \\ \text{Bernoulli}(1/2) & \text{if } s_1 = s_2 \end{cases} \tag{8}$$

In this simplified scheme, at each trial $t$, the two stimuli $s_1^t$ and $s_2^t$ are perfectly perceived with a finite probability $1 - \epsilon$, with $\epsilon < 1$. Under the assumption that the decision maker behaves optimally based on the perceived stimuli, a correct perception would necessarily lead to the correct label. However, with probability $\epsilon$, the first stimulus is randomly selected from a buffer of stimuli, that is, is replaced by a random variable $\hat{s}_1$ that has a probability distribution $p_m^t$.

The probability distribution $p_m^t$ is the statistics of previously shown stimuli. The information about the previous stimulus is given by the activity of the 'slower' PPC network. As shown above, after the presentation of the first stimulus of the trial, the bump of activity is seen to jump to the position encoding one of the previously presented stimuli, $s_2^{t-1}$, $s_1^{t-1}$, $s_2^{t-2}$, etc., with decreasing probability (*Figure 3C*). Therefore, in calculating the performance in the task, we can take $p_m^t$ to be the marginal distribution of the stimulus $s_1$ or $s_2$ across trials, as in the histogram (*Figure 4A*).

The probability of a misclassification is then given by the probability that, given the pair $(s_1^t, s_2^t)$, at trial $t$,

1. the first stimulus is replaced by a random value, which happens with probability $\epsilon$, and
2. the value of $\hat{s}_1$ replaced is larger than $s_2^t$ when $s_1^t$ is smaller and vice versa (*Figure 4C*).

In summary, the probability of an error at trial $t$ is given by

$$\text{Prob}\left\{\text{error} \,\middle|\, s_1^t = s_1, \, s_2^t = s_2\right\} = \epsilon \cdot \begin{cases} p_m^t(s_2)/2 + \sum_{s < s_2} p_m^t(s) & \text{if } s_1 > s_2 \,, \\ p_m^t(s_2)/2 + \sum_{s > s_2} p_m^t(s) & \text{if } s_1 < s_2 \,. \end{cases} \tag{9}$$

## Bayesian description of contraction bias

We reproduce here the theoretical result from *Loewenstein et al., 2021*, which provides a normative model for contraction bias in the Bayesian inference framework, and apply it to the different stimulus

distributions described in section 'The stimulus distribution impacts the pattern of contraction bias through its cumulative'.

A stimulus with value $s$ is encoded by the agent through a noisy representation $\hat{r} \sim \ell(\cdot|s)$. Before the presentation of the stimulus, the agent has an expectation of its possible values which is described by the probability $\pi$. Assuming that it has access to the internal representation $r$, as well as the probability distributions $\ell$ and $\pi$, the agent can infer the perceived stimulus $\hat{s}$ through Bayes rule:

$$p(\hat{s} = s|r) = \frac{\ell(r|s)\,\pi(s)}{p(r)} \tag{10}$$

where $p(r) = \int ds'\, \ell(r|s')\,\pi(s')$. In this Bayesian setting, the probability distributions for the noisy representation and expected measurement are interpreted as the likelihood and the prior, respectively.

In the delayed comparison task, at the time of the decision, the two stimuli $s_1$ and $s_2$ are assumed to be encoded independently, although with different uncertainties, due to the different delays leading to the time of decision: $\ell(r_1, r_2|s_1, s_2) = \ell_1(r_1|s_1)\,\ell_2(r_2|s_2)$, with $\mathrm{var}[\ell_1] > \mathrm{var}[\ell_2]$. Similarly, the expected values of the stimuli are assumed to be independent but also identically distributed: $\pi(s_1, s_2) = \pi(s_1)\,\pi(s_2)$.

The optimal Bayesian decision maker uses the inference of the stimuli through *Equation 10* to produce an estimate of the probability that $s_1 < s_2$, given the internal representations,

$$p(\hat{s}_1 < \hat{s}_2|r_1, r_2) = \iint ds'_1\, ds'_2\, \Theta(s'_2 - s'_1)\, p(s'_1|r_1)\, p(s'_2|r_2) \tag{11}$$

where $\Theta$ is the Heaviside function and yields a label $\hat{y} = 1$ (truth value of '$s_1 < s_2$') when such probability is higher than $1/2$, and $\hat{y} = 0$ otherwise. Therefore, the probability that the Bayesian decision maker yields the response '$s_1 < s_2$' given the true values of the stimuli $s_1$ and $s_2$ are the average of the label $\hat{y}$ over the possible values of their representations, that is, over the likelihood:

$$p(\hat{y} = 1|s_1, s_2) = \iint dr'_1\, dr'_2\; \Theta\left(p(\hat{s}_1 < \hat{s}_2|r'_1, r'_2) - \frac{1}{2}\right) \ell_1(r'_1|s_1)\, \ell_2(r'_2|s_2) \tag{12}$$

## Application to our study

In modeling our data, we assume that the likelihood functions $\ell_1(\cdot|s_1)$ and $\ell_2(\cdot|s_2)$ are Gaussian with mean equal to the stimulus, but with different standard deviations, $\sigma_1$ and $\sigma_2$, respectively, as in *Loewenstein et al., 2021*. We restrict to the particular case where $\sigma_2 = 0$, that is, there is no uncertainty in the representation of the second stimulus, since there is negligible delay between its presentation and the decision. We instead assume a finite standard deviation $\sigma_1 = \sigma$, which we use as the only free parameter of this model to produce *Figure 4—figure supplement 1A–D*, panels 2 and 4.

The prior $\pi$ is chosen to be the marginal distribution of the first stimulus – identical to the marginal of the second stimulus, because of symmetry.

When $\sigma_2 = 0$, $\ell_2(r|s) = \delta(r - s)$ (Dirac delta), and the predicted response probability, *Equation 12*, reduces to

$$p(\hat{y} = 1|s_1, s_2) = \int dr'_1\; \Theta\left(\int_{-\infty}^{s_2} ds'_1 p(s'_1|r_1) - \frac{1}{2}\right) \ell_1(r'_1|s_1). \tag{13}$$

## **Generalized Linear Model (GLM)**

### GLM as in Lieder et al.

Similarly to *Lieder et al., 2019*, we performed a multivariate logistic regression (an instance of GLM) to the output of the network in the delayed discrimination task with recent stimuli values as covariates:

$$P(\text{``s}_1^t < \text{s}_2^t\text{''}\;) = \sigma\left(\alpha\,(s_1^t - s_2^t) + \sum_{i=1}^{h} w_i\,(\overline{s^{t-i}} - s_1^t) + w_{\text{mean}}\,(\langle s \rangle - s_1^t)\right) \tag{14}$$

where $\sigma$ is the sigmoidal function $\sigma(z) = 1/(1 + e^{-z})$, $\overline{s^\tau} = (s_1^\tau + s_2^\tau)/2$ is the mean of the stimuli presented at trial $\tau$, $h$ is the number of 'history' terms in the regression, and $\langle s \rangle$ is the mean of the stimuli within and across trials up to the current one. As in *Lieder et al., 2019*, we choose $h = 4$, that is, we include

in the short-term history the four trials prior to the current one. The first term in *Equation 14*, with weight $\alpha$, controls the slope of the psychometric curve. The remaining terms, combined linearly with weights $w$, contribute to biases expressing the long- and short-term memory. In *Lieder et al., 2019*, it is shown that subjects on the ASD conserve the higher long-term weights, $w_{mean}$, while losing the short-term weights expressed by NT subjects. In contrast, DYS subjects conserve a higher bias from the recent stimuli, $w_1$, while losing the higher long-term weights, also expressed by NT subjects.

In order to gain insight into this regression model in terms of our network, we also performed a linear regression of the bump of activity just before the onset of the second stimulus, denoted $\hat{s}_1^t$, versus the same variables:

$$\hat{s}_1^t = s_1^t + \sum_{i=1}^{h} w_i\,(\overline{s^{t-i}} - s_1^t) + w_{mean}\,(\langle s \rangle - s_1^t) \tag{15}$$

In this case, we see that the weights $w$ in the linear regression for $\hat{s}_1^t$ have the same qualitative behavior as the weights for the bias term in the GLM regression for the performance (not shown). This is expected since the decision-making rule in the network – based on the bump location just before and during the second stimulus, $\hat{s}_1$ and $\hat{s}_2 \simeq s_2^t$, respectively – is deterministic, following $P(\text{"}s_1^t < s_2^t\text{"}) = (s_2^t - \hat{s}_1^t)$. Therefore, the bias term in the GLM performed in *Lieder et al., 2019*, *Equation 14*, corresponds to the displacement of the bump location $\hat{s}_1^t$ with respect to the actual stimulus $s_1^t$, modeled to be linearly dependent on the displacement of previous stimuli from $s_1^t$.

## Regression model with infinite history

In the regression formulas in *Equations 14 and 15*, it is possible to give an interpretation of the parameter $w_{mean}$, that is, the weight of the contribution from the covariate corresponding to the mean of the past stimuli. Let us consider two regression models, one in which, in addition to a regressor corresponding to the mean stimulus, regressors corresponding to the stimulus history are included up to trial $h$, and another in which $h = \infty$, that is, infinitely many past stimuli are included as regressors. In this case, *Equation 15* rewrites

$$\hat{s}_1^t = s_1^t + \sum_{i=1}^{\infty} w_i\,(\overline{s^{t-i}} - s_1^t)\,. \tag{16}$$

If we assume that the weights obtained from the regression have roughly an exponential dependence on time (*Figure 9C and D*), we can write

$$w_i = \gamma\,w_{i-1} = \gamma^i\,w_0\,. \tag{17}$$

By equating *Equations 15 and 16*, we would find that

$$
\begin{aligned}
w_{mean}\,(\langle s \rangle - s_1^t) &= \sum_{i=h+1}^{\infty} w_i\,(\overline{s^{t-i}} - s_1^t) \\
&= w_{i+1} \sum_{j=0}^{\infty} \gamma^j\,(\overline{s^{t-(h+1+j)}} - s_1^t) \\
&= \frac{w_{h+1}}{1-\gamma}\,(\langle s \rangle_\gamma - s_1^t)
\end{aligned}
\tag{18}
$$

where

$$\langle s \rangle_\gamma = \sum_{j=0}^{\infty} g_j\,\overline{s^{t-h-1-j}} \tag{19}$$

that is, an average over the geometric distribution $g_j = (1-\gamma)\,\gamma^j$, from time $t-(h+1)$ backward. Since for $\gamma$ large enough we have $\langle s \rangle_\gamma = \langle s \rangle$, we can identify

$$w_{mean} \propto \frac{w_{h+1}}{1-\gamma}\,. \tag{20}$$

This derivation indicates that the magnitude $w_{mean}$ in the infinite history model, given by *Equation 15*, is a function of the discount factor $\gamma$ as well as the weight of the first trial left out from the finite history regression ($w_{h+1}$). A higher $\gamma$ value, that is, a longer timescale for damping of the weights extending into the stimulus history, yields a higher $w_{mean}$. We can obtain $\gamma$ for each condition (NT, ASD, and DYS) by fitting the weights obtained as a function of trials extending into the history (*Figure 9C and D*). As predicted by *Equation 20*, a larger window for short-term history effects (as in the ASD case relative to NT) yields a larger weight for the covariate corresponding to the mean stimulus. Finally, *Equation 20* also predicts that $w_{mean}$ is proportional to $w_{h+1}$, the number of trials back we consider in the regression, $h$, implying that the number of covariates that we choose to include in the model may greatly affect the results. Both of these predictions are corroborated by plotting directly the value of $w_{mean}$ obtained from the regression (*Figure 9E*).

## Acknowledgements

We are grateful to Loreen Hertäg for helpful comments on our figures and Arash Fassihi for helpful discussions. We also thank Guilhem Ibos for pointing out a typo in our figure legends in a previous version of this manuscript. This work was supported by BBSRC (BB/N013956/1, BB/N019008/1), Wellcome Trust (200790/Z/16/Z), Simons Foundation (564408), EPSRC (EP/R035806/1), Gatsby Charitable Foundation (GAT3755), and Wellcome Trust (219627/Z/19/Z).

## Additional information

### Funding

| Funder | Grant reference number | Author |
|---|---|---|
| Wellcome Trust | 219627/Z/19/Z | Athena Akrami |
| Gatsby Charitable Foundation | GAT3755 | Athena Akrami |
| Biotechnology and Biological Sciences Research Council | BB/N013956/1 | Claudia Clopath |
| Biotechnology and Biological Sciences Research Council | BB/N019008/1 | Claudia Clopath |
| Wellcome Trust | 10.35802/200790 | Claudia Clopath |
| Simons Foundation | 564408 | Claudia Clopath |
| Engineering and Physical Sciences Research Council | EP/R035806/1 | Claudia Clopath |

The funders had no role in study design, data collection and interpretation, or the decision to submit the work for publication. For the purpose of Open Access, the authors have applied a CC BY public copyright license to any Author Accepted Manuscript version arising from this submission.

### Author contributions

Vezha Boboeva, Conceptualization, Data curation, Software, Formal analysis, Validation, Investigation, Visualization, Methodology, Writing – original draft, Writing – review and editing; Alberto Pezzotta, Software, Formal analysis, Methodology, Writing – review and editing; Claudia Clopath, Conceptualization, Supervision, Funding acquisition, Investigation, Writing – original draft, Project administration, Writing – review and editing; Athena Akrami, Conceptualization, Resources, Supervision, Funding acquisition, Investigation, Writing – original draft, Project administration, Writing – review and editing

### Author ORCIDs

Vezha Boboeva (iD) http://orcid.org/0000-0002-2476-8714
Claudia Clopath (iD) https://orcid.org/0000-0003-4507-8648
Athena Akrami (iD) http://orcid.org/0000-0001-5711-0903

## Ethics

The study was approved by the University College London (UCL) Research Ethics Committee [16159/001] (London, UK). Before starting the experiment, participants were provided with an information sheet relevant to the experiment they will be performing and asked to sign an informed written consent. By signing this consent form, the participants consent to freely take part in the study and confirm they understand the information they received. The participants additionally confirm they understand that their participation is voluntary and that they are allowed to stop the experiment at any moment and withdraw the data they provided. Participants consent to allow use of their personal data only for the purpose of scientific research, as determined by applicable law. Identifiable personal data are only available to the researchers and securely stored upon need, while the anonymized version of this data is freely available in a public repository. Once published, the participants' contribution will remain non-identifiable.

Reviewer #1 (Public review): https://doi.org/10.7554/eLife.86725.3.sa1
Reviewer #2 (Public review): https://doi.org/10.7554/eLife.86725.3.sa2
Author response https://doi.org/10.7554/eLife.86725.3.sa3

# Additional files

## Supplementary files
• MDAR checklist

## Data availability

The code used to simulate the network model and analyze the results can be found at https://github.com/vboboeva/ParametricWorkingMemory (copy archived at *Boboeva, 2023*). Data from the human behavioral task and code used to analyze it can be found at https://github.com/vboboeva/ParametricWorkingMemory_Data (copy archived on Zenodo: https://zenodo.org/records/10592611).

The following dataset was generated:

| Author(s) | Year | Dataset title | Dataset URL | Database and Identifier |
| --- | --- | --- | --- | --- |
| Boboeva V, Pezzotta A, Clopath C, Akrami A | 2023 | Parametric Working Memory in humans | https://doi.org/10.5281/zenodo.10592611 | Zenodo, 10.5281/zenodo.10592611 |

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

# Appendix 1

## Parameters

**Appendix 1—table 1.** Simulation parameters, when not explicitly mentioned.
Used to produce *Figures 1–5 and 8*, *Figure 2—figure supplements 1 and 2*, and *Figure 3—figure supplement 2*.

| Parameter | Symbol | Default value |
|---|---|---|
| Number of neurons | $N$ | 2000 |
| Neuronal gain | $\beta$ | 5 |
| Range of excitatory interactions (in units of stimulus space length) | $d_0$ | 0.02 |
| Strength of inhibitory weights | $J_0$ | 0.2 |
| Strength of excitatory weights | $J_e$ | 1 |
| Timescale of neuronal integration in WM net (s) | $\tau^W$ | 0.01 |
| Timescale of neuronal integration in PPC (s) | $\tau^P$ | 0.5 |
| Timescale of neuronal adaptation in PPC (s) | $\tau_\theta^P$ | 7.5 |
| Amplitude of adaptation current in PPC | $D^P$ | 0.3 |
| Amplitude of external inputs | $I_{\text{ext}}$ | 1 |
| Strength of weights from PPC to WM net | $J^{P \to W}$ | 0.5 |
| Duration of stimuli (s) | | 0.4 |
| Delay interval (s) | | [2, 6, 10] |
| Intertrial interval (s) | | 6 |
| Width of box stimulus (in units of stimulus space length) | $\delta s$ | 0.05 |

WM, working memory; PPC, posterior parietal cortex.

.

**Appendix 1—table 2.** Simulation parameters (*Figure 3—figure supplement 1*).

| Parameter | Symbol | Default value |
|---|---|---|
| Number of neurons | $N$ | 1000 |
| Neuronal gain | $\beta$ | 5 |
| Range of excitatory interactions (in units of stimulus space length) | $d_0$ | 0.02 |
| Strength of inhibitory weights | $J_0$ | 0.2 |
| Strength of excitatory weights | $J_e$ | 1 |
| Timescale of neuronal integration (s) | $\tau^P$ | 0.01 |
| Amplitude of external inputs | $I_{\text{ext}}$ | 1 |
| Duration of stimuli (s) | | 0.4 |
| Width of box stimulus (in units of stimulus space length) | $\delta s$ | 0.05 |

**Appendix 1—table 3.** Simulation parameters (*Figure 9* and *Figure 9—figure supplement 1*).
Other parameters as in Table 1*Appendix 1—table 1*.

*Appendix 1—table 3 Continued on next page*

| Parameter | Symbol | Default value |
|---|---|---|
| Timescale of neuronal adaptation in PPC (s) | $\tau_\theta^P$ | 7.5 (DYS), 10 (NT), 15 (ASD) |
| Amplitude of adaptation in PPC | $D^P$ | 0.2 |
| Delay interval (s) | | [2, 6, 10] |
| Intertrial interval (s) | | 2.2 |

ASD, autistic spectrum; DYS, dyslexic; NT, neurotypical; PPC, posterior parietal cortex.

