## [Editor Report · eLife assessment]

This **important** study combines disparate results from both psychophysics and neural silencing experiments to suggest a new interpretation of how animals and humans represent and interpret recent events in our memory. A key aspect of the model put forward here is the presence of discrete jumps in neural activity within the posterior parietal region of the cortex. The model is distinct from other models, and the authors provide **convincing** evidence to support it both from existing results as well as from novel experiments.

---

## [Referee Report · Reviewer #1 (Public review)]

This paper aims to explain recent experimental results that showed deactivating the PPC in rats reduced both the contraction bias and the recent history bias during working memory tasks. The authors propose a two-component attractor model, with a slow PPC area and a faster WM area (perhaps mPFC, but unspecified). Crucially, the PPC memory has slow adaptation that causes it to eventually decay and then suddenly jump to the value of the last stimulus. These discrete jumps lead to an effective sampling of the distribution of stimuli, as opposed to a gradual drift towards the mean that was proposed by other models. Because these jumps are single-trial events, and behavior on single events is binary, various statistical measures are proposed to support this model. To facilitate this comparison, the authors derive a simple probabilistic model that is consistent with both the mechanistic model and behavioral data from humans and rats. The authors show data consistent with model predictions: longer interstimulus intervals (ISIs) increase biases due to a longer effect over the WM, while longer intertrial intervals (ITIs) reduce biases. Finally, they perform new experiments using skewed or bimodal stimulus distributions, in which the new model better fits the data compared to Bayesian models.

The mechanistic proposed model is simple and elegant, and it captures both biases that were previously observed in behavior, and how these are affected by the ISI and ITI (as explained above). Their findings help rethink whether our understanding of contraction bias is correct.

On the other hand, the main proposal - discrete jumps in PPC - is only indirectly verified. The majority of the behavioral predictions stem from the probabilistic model, which is consistent with the mechanistic one, but does not necessitate it.

The revised submission uses the self-paced nature of the experiments to confirm the systematic change in bias with inter-trial-interval, as predicted by the model. This analysis strengthens the hypothesis.

---

## [Referee Report · Reviewer #2 (Public review)]

Working memory is not error free. Behavioral reports of items held in working memory display several types of bias, including contraction bias and serial dependence. Recent work from Akrami and colleagues demonstrates that inactivating rodent PPC reduces both forms of bias, raising the possibility of a common cause.

In the present study, Boboeva, Pezotta, Clopath, and Akrami introduce circuit and descriptive variants of a model in which the contents of working memory can be replaced samples from recent sensory history. This volatility manifests as contraction bias and serial dependence in simulated behavior, parsimoniously explaining both sources of bias. The authors validate their model by showing that it can recapitulate previously published and novel behavioral results in rodents and neurotypical and atypical humans.

Both the modeling and the experimental work is rigorous, providing convincing evidence that a model of working memory in which reports sometimes sample past experience can produce both contraction bias and serial dependence, and that this model is consistent with behavioral observations across rodents and humans in the parametric working memory (PWM) task.

These efforts constitute an admirable initial validation of the proposed model, and the authors present several novel predictions that will allow for further tests in future experiments. First, the authors note that their circuit model predicts a bimodal error distribution in delayed estimation paradigms (due to noisy sampling of sensory history on a subset of trials) that merges into a unimodal distribution when recent sensory history and the current to-be-reported stimulus have very similar values (Fig. 5c). Analysis of extent delayed estimation datasets (e.g., https://osf.io/jmkc9/) or new experiments will provide the opportunity for a straightforward test of this hypothesis.

Second, the bulk of the modeling efforts presented here are devoted to a circuit-level description of how putative posterior parietal cortex (PPC) and working-memory (WM) related networks may interact to produce such volatility and biases in memory. This effort is extremely useful because it allows the model to be constrained by neural observations and manipulations in addition to behavior, and the authors begin this line of inquiry here (by showing that the circuit model can account for effects of optogenetic inactivation of rodent PPC). As the authors note, population electrophysiology in PPC and WM-related areas and single-trial analyses will play an important role in the ultimate validation of this model.

Finally, it is noteworthy that, in the spirit of moving away from an overreliance on p-values (e.g., Amrhein et al., PeerJ 2017), the authors eschew conventional hypothesis testing when reporting their experimental results. The p-values aren't missed, in large part thanks to excellent visualizations and apparently large effect sizes. It's unclear how well this approach would generalize to other questions and datasets; nevertheless, this study provides an interesting data point in the ongoing conversation around reproducibility and rigor.

---

## [Author Response]

We thank the editors and the reviewers for their assessment of our revised manuscript. Please see bellow, our answers to the recommendations by reviewer #2.

Figure S2F - Seems like a very narrow range of parameters. Is there some fine tuning here?

The range of values of tau_P that yields previous-trial biases is bounded by below and above for the following reasons: above a certain value of tau_P (therefore large integration time), the bump that had formed in the previous trial is not strong enough to remain stable for a long time, and therefore dissipates by the time the current trial starts (especially when adaptation is fast, towards the left of the third panel). Below a certain value, instead, this integration timescale is small enough to quickly form a representation of the current trial, hence the bump from the previous trial quickly dissipates (due to mutual inhibition). This interplay between the integration and the adaptation timescale as well as considering a phenomenon which is bounded in time (how close the activity bump is to the second stimulus of the previous trial which is presented between -22.4 and -5.6 seconds from the moment we are considering) yields a region for tau_P which is bounded. This region, however, appears narrow due to the limited number of points we have considered for the simulation grid.

Regarding my comment on lapse at the boundaries (old line 221). Lapse parameters in psychometric curves correspond to errors on the "easy" trials. But the mechanistic explanation for lapse trials is that there is a non-zero probability for the subject to respond in a manner that is random and independent of the stimulus. In the case of extreme stimuli, this is the only reason for errors, and thus looking at the edges of the psychometric curves allows to calculate lapse rate. But - the usual assumption for underlying mechanism is that the subject lapses in all trials, regardless of stimulus. If I understand correctly, this is different than the mechanistic reason for lapses in the network model, which was described as something that happens more in the edges than in the center. Or more generally, to be a stimulus-dependent effect.

We thank the reviewer for this clarification. The reviewer is right that in our mechanistic model, lapses (as defined by errors on easy trials) are more likely to occur for extreme stimuli, due to the vicinity to the boundary of the attractor. Such errors also occur for non-extreme stimuli, when delay intervals are long enough for the bump in PPC to drift to the boundaries. In experiments, lapse trials as described by the reviewer occur due to multiple different reasons; for lapse that is independent of the stimuli, mechanisms such as attention have been thought to play a role, this however is not included in our model.

What are the parameters for the distributions (skewed, bimodal, ...)?

These parameters are reported in the legend of Fig.6, where the distributions appear.

Bump with adaptation. Sorry for the draft-like comment. I don't think the existing studies are in the form you describe. I do think it might be useful to point readers to these studies. If an interested reader wishes to understand network dynamics in this and similar scenarios, it might be useful to have the pointers. The reference I had in mind was Romani, S., & Tsodyks, M. (2015). Short‐term plasticity based network model of place cells dynamics. Hippocampus, 25(1), 94-105.

We thank the reviewer for the clarification, and we will include this reference in the Version of Record.

The following is the authors’ response to the original reviews.

**eLife assessment**
This is an important study about the mechanisms underlying our capacity to represent and hold recent events in our memory and how they are influenced by past experiences. A key aspect of the model put forward here is the presence of discrete jumps in neural activity with the posterior parietal region of the cortex. The strength of evidence is largely solid, with some weaknesses noted in the methodology. Both reviewers suggested ways in which this aspect of the model can to be tested further and resolve conflicts with previously published experimental results, in particular the study by Papadimitriou et al 2014 in Journal of Neurophysiology.

We thank the editors for their assessment. As mentioned in the cover letter, we have addressed all the reviewers’ concerns and would like to request and update of the assessment to reflect the revisions we have made.

**Public Reviews:**

We thank both reviewers for their careful reading and feedback that helped clarify many aspects of the model. Below, we address their comments.

**Reviewer #1 (Public Review):**
This paper aims to explain recent experimental results that showed deactivating the PPC in rats reduced both the contraction bias and the recent history bias during working memory tasks. The authors propose a twocomponent attractor model, with a slow PPC area and a faster WM area (perhaps mPFC, but unspecified). Crucially, the PPC memory has slow adaptation that causes it to eventually decay and then suddenly jump to the value of the last stimulus. These discrete jumps lead to an effective sampling of the distribution of stimuli, as opposed to a gradual drift towards the mean that was proposed by other models. Because these jumps are single-trial events, and behavior on single events is binary, various statistical measures are proposed to support this model. To facilitate this comparison, the authors derive a simple probabilistic model that is consistent with both the mechanistic model and behavioral data from humans and rats. The authors show data consistent with model predictions: longer interstimulus intervals (ISIs) increase biases due to a longer effect over the WM, while longer intertrial intervals (ITIs) reduce biases. Finally, they perform new experiments using skewed or bimodal stimulus distributions, in which the new model better fits the data compared to Bayesian models.The mechanistic proposed model is simple and elegant, and it captures both biases that were previously observed in behavior, and how these are affected by the ISI and ITI (as explained above). Their findings help rethink whether our understanding of contraction bias is correct.On the other hand, the main proposal - discrete jumps in PPC - is only indirectly verified.

We agree with the reviewer that the evidence for discrete jumps in PPC has been provided in behavioural results (short-term, n-back trial biases), and not from neural data. However, we believe electrophysiological investigations are out of the scope of the current manuscript and future works are needed to further verify the results.

The model predicts a systematic change in bias with inter-trial-interval. Unless I missed it, this is not shown in the experimental data. Perhaps the self-paced nature of the experiments allows to test this?

We thank the reviewer for this great suggestion.

We had not previously looked at this in the data for the reason that in the simulations, the ITI is set to either 2.2, 6 or 11 seconds, whereas the experiment is self-paced. Therefore, any comparison with the simulation should be made carefully.

However, after the reviewer’s suggestion, we did look at the change in the bias with the inter-trial interval, by dividing trials according to ITIs lower than 3 seconds (“short” ITI), and higher than 3 seconds (“long” ITI). This choice was motivated by the shape of the distribution of ITIs, which is bimodal, with a peak around 1 second, and another after 3 seconds (new Fig 8F). Hence, we chose 3 seconds as it seemed a natural division. However, 3 seconds also happens to be approximately the 75th percentile of the distribution, and this means that there is much more data in the “short” ITI than the “long” ITI set. In order to have sufficient data in the “long” ITI for clearer effects we used all of our dataset – the negatively skewed, and also two bimodal distributions (of which only one was shown in the manuscript, for succinctness). This larger dataset allows us to clearly see not only a decreasing contraction bias with increasing ITI (Fig 8G), but also a decreasing onetrial-back attractive bias with increasing ITI (Fig 8H). We have uploaded all the datasets as well as scripts used to analyze them to this repository:https://github.com/vboboeva/ParametricWorkingMemory_Data.

The data in some of the figures in the paper are hard to read. For instance, Figure 3B might be easier to understand if only the first 20 trials or so are shown with larger spacing. Likewise, Figure 5C contains many overlapping curves that are hard to make out.

We have limited the dynamics in Fig 3B to the first 50 trials for better visibility. Likewise, as suggested, we report the standard error of the mean instead of the standard deviation in old Fig 5C (new Fig 6C) – this allows for the different curves to be better discernible.

There is a gap between the values of tau_PPC and tau_WM. First - is this consistent with reports of slower timescales in PFC compared to other areas?

Recent studies by Xiao-Jing Wang and colleagues (Refs. 1-3 below) suggest that may be the case. In Wang et al 2023, Ref 1 below, the authors use a generative model to study the concept of bifurcation in space in working memory, that is accompanied by an inverted-V shape of the time constants as a function of cortical hierarchy.

Briefly, they propose a generative model of the cortex with modularity, incorporating repeats of a canonical local circuit connected via long-range connections. In particular, the authors define a hierarchy for each local circuit. At a critical point in this hierarchy axis, there is a phase transition from monostability to bistability in the firing rate. This means that a local circuit situated below the critical point will only display a low activity steady state, while those above the critical point additionally display a persistent activity steady state.

The model predicts a critical slowing down of the neural fluctuations at the critical point, resulting in an inverted-V shape of the time constants as a function of the hierarchy. They test the predictions of their model – the bifurcation in space and that inverted-V-shaped time constants as a function of the hierarchy - on connectome-based models of the macaque and mouse cortex. Interestingly both datasets show similar behavior. In particular, during working memory, frontal areas (higher in the hierarchy, e.g. area 24c in macaques) has a smaller time constant relative to posterior parietal areas (lower in the hierarchy, like LIP or f7). We have now cited this new work.

[1] https://www.biorxiv.org/content/10.1101/2023.06.04.543639v1

[2] https://elifesciences.org/articles/72136

[3] https://www.biorxiv.org/content/10.1101/2022.12.05.519094v3.abstract

Second - is it important for the model, or is it mostly the adaptation timescale in PPC that matters?

We have run simulations producing a phase diagram with tau_theta^P on the x-axis, tau^P on the y-axis, and in color, the fraction of trials in which the bump is in the vicinity of a target (Fig S2 F), before the network is presented with the second stimulus. This target can be the first stimulus s_1 (left), mean over stimuli (middle) and previous trial’s stimulus (right). White point corresponds to parameters of the default network.

In this phase diagram, the lowest value that tau_P takes is tau_WM=0.01. When tau_P=tau_WM, the bump is rarely in the vicinity of 1-trial-back stimulus, and we can see that tau_PPC should be greater than tau_WM in order for the model to yield 1-trial back effects. We conclude that it is indeed important for tau_PPC > tau_WM.

We have included this in Fig S2 F of the manuscript.

Regarding the relation to other models, the model by Hachen et al (Ref 45) also has two interacting memory systems. It could be useful to better state the connection, if it exists.

The model proposed by Hachen et al is conceptually different in that one module stores the mean of the sensory stimulus; it could be related to a variant of our model where adaptation is turned off in the PPC network (Fig S2 A). However, the task they model is also different: subjects have to learn the location of a boundary according to which the stimulus is classified as ‘weak’ or ‘strong’, set by the experimenter. Hence, it is a task where learning is needed - this contrasts with the task we are modelling, where only working memory is required. How task demands reconfigure existing circuits via dynamics and/or learning to perform different computations is a fascinating area of research that is outside the scope of this work.

**Reviewer #2 (Public Review):**
Working memory is not error free. Behavioral reports of items held in working memory display several types of bias, including contraction bias and serial dependence. Recent work from Akrami and colleagues demonstrates that inactivating rodent PPC reduces both forms of bias, raising the possibility of a common cause.In the present study, Boboeva, Pezzotta, Clopath, and Akrami introduce circuit and descriptive variants of a model in which the contents of working memory can be replaced by previously remembered items. This volatility manifests as contraction bias and serial dependence in simulated behavior, parsimoniously explaining both sources of bias. The authors validate their model by showing that it can recapitulate previously published and novel behavioral results in rodents and neurotypical and atypical humans.Both the modeling and the experimental work is rigorous, providing compelling evidence that a model of working memory in which reports sometimes sample past experience can produce both contraction bias and serial dependence, and that this model is consistent with behavioral observations across rodents and humans in the parametric working memory (PWM) task.Evidence for the model advanced by the authors, however, remains incomplete. The model makes several bold predictions about behavior and neural activity, untested here, that either conflict with previous findings or have yet to be reported but are necessary to appropriately constrain the model.First, in the most general (descriptive) formulation of the Boboeva et al. model, on a fraction of trials items in working memory are replaced by items observed on previous trials. In delayed estimation paradigms, which allow a more direct behavioral readout of memory items on a trial-by-trial basis than the PWM task considered here, reports should therefore be locked to previous items on a fraction of trials rather than display a small but consistent bias towards previous items. However, the latter has been reported (e.g., in primate spatial working memory, Papadimitriou et al., J Neurophysiol 2014). The ready availability of delayed estimation datasets online (e.g., from Rademaker and colleagues, https://osf.io/jmkc9/) will facilitate in-depth investigation and reconciliation of this issue.

As pointed out by the reviewer, in the PWM task that we are modelling here, the activity in the network is used to make a binary decision. However, it is possible to directly analyse the network activity before the onset of the second stimulus.

In their manuscript, Papadimitriou et al. study a memory-guided saccade task in nonhuman primates and argue that the animals display a small but consistent bias towards previous items (Fig 2). In that figure, the authors compute the error as the difference between the saccade direction and target direction in each trial. They compute this error for all trials in which the preceding trial’s target direction is between 35° and 85° relative to the current trial (counterclockwise with respect to the current trial’s target). They discover that the residual error distribution is unimodal with a mode at 1.29° and a mean at 2.21° (positive, so towards the preceding target’s direction), from which they deduce a small but systematic bias towards previous trial targets.

We have computed a similar measure for our network with default parameters (Table 1), by subtracting the location of the bump at the end of the delay interval (s_hat(t), ‘saccade’) from the initial location of the first stimulus in the current trial (s1(t) or the ‘target’). We have done this for all trials where s1(t)=0.2, and where s2(t-1) takes specific values. These distributions are characterized by two modes. The first corresponds to those trials where the bump is not displaced in WM (i.e. mean of zero). We can also see the appearance of a second mode at the location of s1(t) - s2(t-1), corresponding to the displacements towards the preceding trial’s stimulus described in the main text. If, instead, we limit the analysis to a small range of previous trials close to s1(t) (similar to Papadimitriou et al) then the distribution of residual errors will appear unimodal, as the two modes merge. Importantly, note that there is a large variability around the second mode, expressing a more complex dynamics in the network. As can be seen in Fig 3B, the location of the bump is not always slaved to the one in the PPC in a straightforward way -- due to the adaptation in the PPC, the global inhibition in the connectivity kernel, as well as interleaved design for various delay intervals, the WM bump can be displaced in nontrivial ways (see also Recommendation no 4), yielding the dispersion around the second peak. It remains to be seen whether such patterns can be observed in the data from previous works on continuous working memory recall (including Papadimitriou et al). However, to our knowledge, such detailed and full analysis of errors at the level of individual trials has not been done.

In summary, this analysis shows that the type of dynamics in our network is not one of the two cases: (1) small and systematic bias in each and every trial or (2) large error that occurs only rarely; rather, the dispersion around both modes suggests that the dynamics in our model are a mixture of these two limit cases.

We have also performed another typical analysis, reported in several continuous recall tasks (e.g. Jazayeri and Shadlen 2010) where contraction bias has been reported. We plot WM bump locations after the delay period for every trial (s_hat(t)), and their averages, against the nominal value of s1(t). We see that the mean WM location deviates from the identity line toward the mean values of s1(t), again showing contraction bias as an average effect, while individual trials follow the dynamics explained above.

We have now included a new section on continuous recall (Sect. 1.5 and a new figure (Fig 5)), which details the two above-mentioned analyses. The analysis of freely available datasets of delayed estimation tasks, unfortunately, is out of the scope of this work, and we leave such analyses to future studies.

Second, the bulk of the modeling efforts presented here are devoted to a circuit-level description of how putative posterior parietal cortex (PPC) and working-memory (WM) related networks may interact to produce such volatility and biases in memory. This effort is extremely useful because it allows the model to be constrained by neural observations and manipulations in addition to behavior, and the authors begin this line of inquiry here (by showing that the circuit model can account for effects of optogenetic inactivation of rodent PPC).Further experiments, particularly electrophysiology in PPC and WM-related areas, will allow further validation of the circuit model. For example, the model makes the strong prediction that WM-related activity should display 'jumps' to states reflecting previously presented items on some trials. This hypothesis is readily testable using modern high-density recording techniques and single-trial analyses.

As mentioned in response to the previous comment, we note again that in the WM network, the bump ‘displacement’ has a complex dynamics -- the examples we have provided in Fig 1A and 2B mainly show the cases in which jumps occur in the WM network, but this is not the only type of dynamics we observe in the model. We do have instances in which the continuity of the model causes drift across values, and we have now replaced the right panel in Fig 2B with one such instance, in order to emphasize that this displacement towards the previous trial’s stimulus (s2(t-1)) can occur in various ways. For a more thorough analysis, we have analyzed the distance between s1(t) and the position of the bump in the WM network at the end of the delay period s_hat(t), conditioned on specific values of s1(t) and s2(t-1) (Fig 5C). In this figure, we can see the appearance of two modes: one centered around 0, corresponding to the correct trials where the stimulus is kept in WM (s1(t) = s_hat(t)), and another mode centered around s2(t-1), the location of the second stimulus of the previous trial, where the bump is displaced. Note, as we explain in Sect. 1.5, the large dispersion around this second mode, which suggests that the bump is not always displaced to that specific location and may undergo drift.

We agree with the reviewer that future electrophysiological experiments (or analysis of existing datasets) are necessary for validation of these results.

Finally, while there has been a refreshing movement away from an overreliance on p-values in recent years (e.g., Amrhein et al., PeerJ 2017), hypothesis testing, when used appropriately, provides the reader with useful information about the amount of variability in experimental datasets. While the excellent visualizations and apparently strong effect sizes in the paper mitigate the need for p-values to an extent, the paucity of statistical analysis does impede interpretation of a number of panels in the paper (e.g., the results for the negatively skewed distribution in 5D, the reliability of the attractive effects in 6a/b for 2- and 3- trials back).

We share the reviewer’s criticism towards the misuse of p-values – in order for a clearer interpretation of old Fig 5D (new Fig 7E), we have looked at the 2 and 3 trials-back biases by using all of our dataset – the negatively skewed, and also two bimodal distributions (of which only one was shown in the manuscript). This larger dataset of 43 subjects (approximately 17,200 trials) allows us to clearly see the 2 and 3 trial back attractive biases, and the effect that the delay interval exerts on them.

**Reviewer #1 (Recommendations For The Authors):**
Fig 5 A&C - It might be beneficial to separate the distribution of stimuli from the performance. It is hard to read the details of the performance, especially with error bars.

Following the next recommendation, we have exchanged the standard deviation to standard errors of the mean, hopefully this allows to better read the performance.

Fig 5C. The number of participants should be written. Perhaps standard errors instead of standard deviation?

We have now changed the standard deviation to standard errors of the mean and included the number of participants in the figure.

Fig 2B - hard to understand, because there is no marking of where "perfect" memory of s1 would be.

The perfect memory of s1 is shown in the upper panel as black bars.

Fig 3B. dot number 9 (blue, around 0.7) - why is WM higher than stimulus?

This trial has a long ISI (blue means 10s). During this delay, the bump in the PPC, under the influence of adaptation, drifts far below the first stimulus (note that the previous trial also had its first stimulus in the same location, as a result of which the adaptative thresholds have built up significantly, causing the bump to move away from that location). During this delay period, neurons in the WM network receive inputs from the PPC network: if this input is strong enough, it can disrupt an existing bump; if not, this input still exerts inhibiting influence on the existing bump via the global inhibition in the connectivity. This can cause an existing bump to slowly drift in a random direction, and finally dissipate. Note that the lines in Fig 2B represent the neuron with the maximal activity, this activity may be a stable bump, or an unstable bump that may soon dissipate.

Other examples with similar dynamics include trials 43 and 54.

L167 fewer -> smaller

We have now corrected this.

Fig 3C - bump can also be in between. Is this binned?

We have not binned the length of the attractor; to produce that figure, we check whether the position of the neuron with the maximal firing rate is within a distance of ±5% of the length of the whole line attractor from the target location.

L221 Lapse at the boundary of attractor. This seems very different from behavior.Specifically, if it is in the boundaries, it should be stimulus dependent.

Very sorry, we did not manage to understand the reviewer’s comment.

L236 are -> is

We have now corrected this.

Fig S4 - should be mostly in main text.

Part of this figure is in Fig 6A, but given the amount of detail, we think Supplementary Material is better suited.

L253-254. Differences across all distributions - very minor except the bimodal case.

That is correct, this is why we conducted the experiment with the bimodal distribution, to better differentiate the predictions of the two models.

L273 extra comma after "This probability"

We have now corrected this.

ITI was only introduced in section 1.5.2. Perhaps worth mentioning the default 5s value earlier in the paper.

We have now mentioned this in line 97-98.

Fig S6B title: perhaps "previous stimuli"?

We have now corrected this.

L364 i"n A given trial"Equation 2 - no decay term?

Thank you for pointing out this error, we have now corrected this.

Equation 5,6 are j^W and j^P indices of neurons in those populations?

Yes, j^W indexes neurons in the WM network, and j^P those in the PPC. We have now added this in the text for clarity.

Bump with adaptation - other REFs? Sandro?

We are aware of continuous bump attractors implementing short-term synaptic plasticity in various studies (including by Sandro Romani), but not in the form we have described. May the reviewer kindly point us towards the relevant literature.

Free boundary - what is the connectivity for neurons 1 and N? Is it weaker than others? Is the integral still 1? Does this induce some bias on the extreme values?

The connectivity of the network is all-to-all. However, as expressed by Eq. (3), the distance-dependent contribution to the weights, K, decreases exponentially as we move from neuron 1 onwards, and from neuron N down. The sum (or integral, in the large-N limit) of the K_ij for j on either side of neuron i is unity only when i is sufficiently far from 1 or N. We have rephrased the paragraph starting in line 516 to make this clearer.

The presence of a boundary could introduce a bias in theory, but in practice, it affects the dynamics only when the bump drifts sufficiently close to it. The smallest stimulus in the simulated task has amplitude 0.2, with width 0.05, which implies the activation of 50 neurons on either side of neuron 400. If one compares this with the width of the kernel K in stimulus space (d_0 = 0.02), which spans ~10 neurons, we can see that the bump of activity stays mostly far from the boundary. It is possible, though it is observed rarely, when several consecutive long delay intervals happen to occur, that the bump in PPC drifts beyond the location corresponding to either the minimum or maximum stimulus.

Code availability?

Code simulating the dynamics of the network as well as analysing the resulting data can be found in the following repository: https://github.com/vboboeva/ParametricWorkingMemoryCode used to analyse human behavioural data and fit them with our statistical model can be found in this repository: https://github.com/vboboeva/ParametricWorkingMemory_DataCode used to run the auditory PWM experiments with human subjects (adapted from Akrami et al 2018) can be found here: https://github.com/vboboeva/Auditory_PWM_human

L547 stimuli

We have now corrected this.

Equation 14 uses both stimuli. Was this the same for the rest of analysis in the paper (first figures for instance)?

This equation was used for all GLM analyses (Figs 9 and S6).

D0 is very small (0.02). Does this mean that activity is essentially discrete in the model? Fig 1A & 2B - the two examples of model activity suggest this is the case. In other words - are there cases where the continuity of the model causes drift across values? Can you show an example (similar to Fig 1A)?

Since this point has been raised beforehand, we refer to the first comment, Fig 2B and Sect. 1.5 for the response to this question.

Table 1 - inter trial interval 6. Text says 5

We have now corrected this in the text.

**Reviewer #2 (Recommendations For The Authors):**
In addition to my review above, I just have a few minor comments:If I understood correctly, the squares inside the purple rectangle in Figure 1B are meant to show a gradation from red to blue, but this was hard to make out in the pdf.

Actually the squares are all on one side or the other of the diagonal, therefore they do not have any gradation.

line 164: "The resulting dynamics... [are]?"

We have corrected this in the text.

Fig 7B legend: "The network performance is on average worse for longer ITIs" – correct?

This was a mistake, we have replaced worse with better.

Other comments

We realized that the colorbar reported the incorrect fraction classified in Figs 1B, 2C, 7B (new 8B), S2C, S3A, S5B. We have corrected this in the new version of the manuscript.

We also found a minor mistake in one of our analysis codes that computed the n-trial back biases for different delay intervals. This did not change our results, actually made the effects clearer. The figures concerned are Fig 3F and new Fig 7E.